# An $\alpha$-No-Regret Algorithm For Graphical Bilinear Bandits

**Geovani Rizk**
PSL - Université Paris Dauphine,
CNRS, LAMSADE, Paris, France
geovani.rizk@dauphine.psl.eu

**Igor Colin**
Huawei Noah's Ark Lab,
Paris, France
igor.colin@huawei.com

**Albert Thomas**
Huawei Noah's Ark Lab,
Paris, France
albert.thomas@huawei.com

**Rida Laraki**
PSL - Université Paris Dauphine,
CNRS, LAMSADE, Paris, France
rida.laraki@lamsade.dauphine.fr

**Yann Chevaleyre**
PSL - Université Paris Dauphine,
CNRS, LAMSADE, Paris, France
yann.chevaleyre@lamsade.dauphine.fr

## Abstract

We propose the first regret-based approach to the *Graphical Bilinear Bandits* problem, where $n$ agents in a graph play a stochastic bilinear bandit game with each of their neighbors. This setting reveals a combinatorial NP-hard problem that prevents the use of any existing regret-based algorithm in the (bi-)linear bandit literature. In this paper, we fill this gap and present the first regret-based algorithm for graphical bilinear bandits using the principle of optimism in the face of uncertainty. Theoretical analysis of this new method yields an upper bound of $\tilde{O}(\sqrt{T})$ on the $\alpha$-regret and evidences the impact of the graph structure on the rate of convergence. Finally, we show through various experiments the validity of our approach.

## 1 Introduction

In this paper, we are interested in solving centralized multi-agent problems that involve interactions between agents. For instance, consider the problem of the configuration of antennas in a wireless cellular network [Siomina et al., 2006] or the adjustment of turbine blades in a wind farm [van Dijk et al., 2016, Bargiacchi et al., 2018]. Choosing a parameter for an antenna (respectively adjusting a turbine blade) has an impact on its signal quality (respectively its energy collection efficiency) but also on the one of its neighboring antennas due to signal interference (respectively its neighboring turbine blade due to wind turbulence). By considering each antenna or turbine blade as an agent, these problems can be modeled as a multi-agent multi-armed bandit problem [Bargiacchi et al., 2018] with the knowledge of a coordination graph [Guestrin et al., 2002] where each node represents an agent and each edge represents an interaction between two agents. As a natural extension of the well-studied linear bandit to this multi-agent setting we focus on the *graphical bilinear bandit* setting introduced in [Rizk et al. [2021]: at each round, a central entity chooses an action for each agent and observes a bilinear reward for each couple of neighbors in the graph that is function of the actions chosen by the two neighboring agents.

36th Conference on Neural Information Processing Systems (NeurIPS 2022).

The objective of the learner can either be to maximize the cumulative reward, which requires a trade-off between *exploration* and *exploitation*, or to identify the best action without being concerned about suboptimal rewards obtained during the learning process, known as the *pure exploration* setting [Bubeck et al., 2009, Audibert and Bubeck, 2010]. For graphical bilinear bandits, while the latter has already been addressed in Rizk et al. [2021], maximizing the cumulative reward (and thus minimizing the associated regret) remains an open question that we address in this paper.

**Contribution.** Given the NP-Hardness property of the problem and the impossibility of designing an algorithm that has sublinear regret with respect to the horizon $T$, we propose the first regret-based algorithm for the graphical bilinear bandit problem that gives a guarantee on the $\alpha$-regret of $\tilde{O}(\sqrt{T})$ where $\alpha$ (greater than $1/2$) is a problem-dependent parameter and $\tilde{O}$ hides the logarithmic factors. The problem as well as the notion of $\alpha$-regret are introduced in Section 2. We provide a first algorithm and establish theoretical guarantees in Section 3. In Section 4, we identify the problem-dependent parameters that one could use to improve both theoretical guarantees and empirical performances of the algorithm. Accordingly, we state a refined bound that significantly improves $\alpha$. Finally, Section 5 gives an experimental overview of the various theoretical bounds established for the regret.

**Related work.** Centralized multi-agent bandit problems where the learner has to choose the actions of all the agents at each round implies to deal with parallelizing the learning process on the agents. In the context of linear rewards where all the agents share the same reward function, Chan et al. [2021] give a detailed analysis of the problem and show that a sublinear regret in $T$ can be reached but with an additional cost specific to the parallelization. The main difference with their work is that they consider independent agents, whereas we assume interactions between the agents.

Choosing an action for each node of a graph to then observe a global reward over the graph is a combinatorial bandit problem [Cesa-Bianchi and Lugosi, 2012], the number of joint arms scaling exponentially with the number of nodes. This has been studied in the regret-based context [Chen et al., 2013, Perrault et al., 2020] where the notion of $\alpha$-regret is introduced to cope with the NP-Hardness of the problem. Most papers assume the availability of an oracle to solve the inner combinatorial optimization problem at each round. In this paper, we do not make this assumption and we present an algorithm that takes into account the graph structure, which has not been considered in the combinatorial framework.

Bandit problems in graphs have been studied from several angles. For instance, on the one hand, in Valko et al. [2014], Mannor and Shamir [2011], each node of the graph is an arm of the bandit problem, which, if pulled, gives information on its neighboring arms. On the other hand, in Cesa-Bianchi et al. [2013] each node is an instance of a linear bandit and the neighboring nodes are assumed to have similar unknown regression coefficients. In this setting, the reward observed for each agent only depends on its chosen action which does not take into consideration agent interactions.

The graphical bilinear bandit setting has been first established in Rizk et al. [2021] where they tackle the problem of identifying the $1/2$-best joint arm (which has an associated global reward equal to at least $1/2$ of the optimal one). They study the pure exploration setting while we tackle the regret-based one. To the best of our knowledge, our paper is the first one to provide regret-based algorithms for the graphical bilinear bandits setting.

## 2 Problem setting

Let $\mathcal{G} = (V, E)$ be the directed graph defined by $V$ the finite set of nodes representing the agents and $E$ the set of edges representing the agent interactions. We assume that if $(i, j) \in E$ then $(j, i) \in E$.[1] The Graphical Bilinear Bandit with a graph $\mathcal{G}$ consists in the following sequential problem [Rizk et al., 2021]: at each round $t$, each agent $i \in V$ chooses an arm $x_t^{(i)}$ in a finite arm set $\mathcal{X}$ and receives for each of its neighbors $j$ a bilinear reward

$$y_t^{(i,j)} = x_t^{(i)\top} \mathbf{M}_\star x_t^{(j)} + \eta_t^{(i,j)} \ , \tag{1}$$

---

[1]One could define an undirected graph instead. However, we consider that interactions between two neighbors are not necessarily symmetrical, with respect to the obtained rewards, so we choose to keep the directed graph to emphasize this asymmetry.

where $\mathbf{M}_\star \in \mathbb{R}^{d \times d}$ is an unknown matrix, and $\eta_t^{(i,j)}$ a zero-mean $\sigma$-sub-Gaussian random variable.[2]

For all agent $i \in V$, we denote $\mathcal{N}_i$ the set of its neighboring agents. Let $n = \mathrm{Card}\,(V)$ denote the number of nodes, $m = \mathrm{Card}\,(E)$ the number of edges and $K = \mathrm{Card}\,(\mathcal{X})$ the number of arms. We assume that for all $x \in \mathcal{X}, \|x\|_2 \leq L$. Let $\|\mathbf{M}_\star\|_F$ be the Frobenius norm of $\mathbf{M}_\star$ with $\|\mathbf{M}_\star\|_F \leq S$. We assume that the expected reward of each edge is positive which gives for all $(x, x') \in \mathcal{X}^2,\ 0 \leq x^\top \mathbf{M}_\star x' \leq LS$.

For each round $t$, we define the global reward $y_t = \sum_{(i,j) \in E} y_t^{(i,j)}$ as the sum of the local rewards obtained over the graph at round $t$, and $\left(x_\star^{(1)}, \ldots, x_\star^{(n)}\right) = \arg\max_{\left(x^{(1)}, \ldots, x^{(n)}\right)} \sum_{(i,j) \in E} x^{(i)\top} \mathbf{M}_\star x^{(j)}$ as the optimal joint arm that maximizes the expected global reward. The corresponding global pseudo-regret over $T$ rounds is then defined as follows:

$$R(T) = \sum_{t=1}^T \left[ \sum_{(i,j) \in E} x_\star^{(i)\top} \mathbf{M}_\star x_\star^{(j)} - \sum_{(i,j) \in E} x_t^{(i)\top} \mathbf{M}_\star x_t^{(j)} \right] \ .$$

We know from Rizk et al. [2021] that finding $\left(x_\star^{(1)}, \ldots, x_\star^{(n)}\right)$ is NP-Hard with respect to the number of agents $n$.

**Proposition 2.1** (Rizk et al. [2021] Theorem 4.1)**.** *Consider a given matrix $\mathbf{M}_\star \in \mathbb{R}^{d \times d}$ and a finite arm set $\mathcal{X} \subset \mathbb{R}^d$. Unless P=NP, there is no polynomial time algorithm guaranteed to find the optimal solution of*

$$\max_{\left(x^{(1)}, \ldots, x^{(n)}\right) \in \mathcal{X}^n} \sum_{(i,j) \in E} x^{(i)\top} \mathbf{M}_\star x^{(j)} \ .$$

Hence, the objective of designing an algorithm with a sublinear regret in $T$ is not feasible in polynomial time with respect to $n$. However, some NP-hard problems are $\alpha$-approximable (for some $\alpha \in (0, 1]$), which means that there exists a polynomial-time algorithm guaranteed to produce solutions whose values are at least $\alpha$ times the value of the optimal solution. For these problems, it makes sense to consider the $\alpha$-pseudo-regret as in [Chen et al., 2013] which is defined for all $\alpha \in (0, 1]$ as follows

$$R_\alpha(T) \triangleq \sum_{t=1}^T \left[ \sum_{(i,j) \in E} \alpha x_\star^{(i)\top} \mathbf{M}_\star x_\star^{(j)} - \sum_{(i,j) \in E} x_t^{(i)\top} \mathbf{M}_\star x_t^{(j)} \right] \ ,$$

and set the objective of designing an algorithm with a sublinear $\alpha$-regret.

**The linear form.** The reward can either be viewed as a bilinear reward as shown in (1) or as a linear reward in higher dimension [Jun et al., 2019] with $y_t^{(i,j)} = \left\langle \mathrm{vec}\left(x_t^{(i)} x_t^{(j)\top}\right), \mathrm{vec}\left(\mathbf{M}_\star\right) \right\rangle + \eta_t^{(i,j)}$ where for any matrix $\mathbf{A} \in \mathbb{R}^{d \times d}$, $\mathrm{vec}\,(\mathbf{A})$ denotes the vector in $\mathbb{R}^{d^2}$ which is the concatenation of all the columns of A, and $\langle \cdot, \cdot \rangle$ refers to the dot product between two vectors.

To simplify the notation, let us refer to any $x \in \mathcal{X}$ as a *node-arm* and define the arm set $\mathcal{Z} = \left\{ \mathrm{vec}\left(xx'^\top\right) | (x, x') \in \mathcal{X} \right\}$ where any $z \in Z$ will be referred as an *edge-arm*. If the arm $x_t^{(i)} \in \mathcal{X}$ represents the node-arm allocated to the node $i \in V$ at time $t$, for each edge $(i,j) \in E$ we will denote the associated edge-arm by $z_t^{(i,j)} := \mathrm{vec}\left(x_t^{(i)} x_t^{(j)\top}\right) \in \mathcal{Z}$ and define $\theta_\star = \mathrm{vec}\,(\mathbf{M}_\star)$ the vectorized

---

[2]One may wish that the agent's reward $i$ also takes into account the quality of its chosen arm alone (*i.e.,* not compared to the one chosen by its neighbors). We discuss this case in the appendix and show that if we add a linear term as a function of $x_t^{(i)}$ to the bilinear reward, the new reward can still be reduced to a simple bilinear reward with an unknown matrix $\mathbf{M}_\star \in \mathbb{R}^{(d+1) \times (d+1)}$.

version of the unknown matrix $\mathbf{M}_\star$ with $\|\theta_\star\|_2 \leq S$. With those notations, the (now) linear reward can be rewritten as follows:

$$y_t^{(i,j)} = \left\langle z_t^{(i,j)}, \theta_\star \right\rangle + \eta_t^{(i,j)} \ . \tag{2}$$

When needed, for any node-arms $(x, x') \in \mathcal{X}^2$ we use the abbreviation $z_{xx'} \triangleq \mathrm{vec}(xx'^\top)$. Finally for any vector $x \in \mathbb{R}^d$ and a symmetric positive-definite matrix $\mathbf{A} \in \mathbb{R}^{d \times d}$, we define $\|x\|_\mathbf{A} \triangleq \sqrt{x^\top \mathbf{A} x}$.

## 3 Optimism in the face of uncertainty for Graphical Bilinear Bandit

Let us first notice that given the linear reward (2) that can be observed for each of the $m$ edges $(i, j) \in E$, the central entity is trying to solve $m$ *parallel* and *dependent* linear bandit problems. The parallel aspect comes from the fact that at each round $t$ the learner has to allocate all the $m$ edge-arms of the graph (one for each edge $(i, j)$ in $E$) before receiving the corresponding $m$ linear rewards. The dependent aspect comes from the fact that the choice of most of the edge-arms are depending on other edge-arms. Indeed, two adjacency edges $(i, j)$ and $(j, k)$ share the node $j$, hence their allocated edge-arms must share the node-arm $x_t^{(j)}$.

In this paper, we choose to design an algorithm based on the principle of optimism in the face of uncertainty [Auer et al.] [2002], and in the case of a linear reward [Li et al.] [2010], [Abbasi-Yadkori et al.] [2011], we need to maintain an estimator of the true parameter $\theta_\star$. To do so, let us define for all rounds $t \in \{1, \ldots, T\}$ the OLS estimator of $\theta_\star$ as follows:

$$\hat{\theta}_t = \mathbf{A}_t^{-1} b_t \ , \tag{3}$$

where,

$$\mathbf{A}_t = \lambda \mathbf{I}_{d^2} + \sum_{s=1}^t \sum_{(i,j) \in E} z_s^{(i,j)} z_s^{(i,j)\top} \ ,$$

with $\lambda > 0$ a regularization parameter and

$$b_t = \sum_{s=1}^t \sum_{(i,j) \in E} z_s^{(i,j)} y_s^{(i,j)} \ .$$

We define also the confidence set

$$C_t(\delta) = \left\{ \theta : \|\theta - \hat{\theta}_t\|_{A_t^{-1}} \leq \sigma \sqrt{d^2 \log \left( \frac{1 + tmL^2/\lambda}{\delta} \right)} + \sqrt{\lambda} S \right\} \ ,$$

where with probability $1 - \delta$, we have that $\theta_\star \in C_t(\delta)$ for all $t \in \{1, \ldots, T\}$, and $\delta \in (0, 1]$.

Since the graphical bilinear bandit can be seen as a linear bandit in dimension $d^2$ and with $K^n$ possible arms,[3] the idea of our method is to overcome the NP-Hard optimization problem that one would have to solve at each round when directly applying the OFUL algorithm as defined in [Abbasi-Yadkori et al.] [2011]. Indeed, at each round, instead of looking for the best optimistic joint arm

$$(x_t^{(1)}, \ldots, x_t^{(n)}) = \underset{(x^{(1)}, \ldots, x^{(n)}) \in \mathcal{X}^n}{\arg\max} \ \underset{\theta \in C_{t-1}(\delta)}{\max} \left\langle \sum_{(i,j) \in E} z_{x^{(i)} x^{(j)}}, \theta \right\rangle \ ,$$

---

[3]It can be seen as a linear bandit problem because the goal of the central entity is to maximize at each round $t$ the expected global reward $\sum_{(i,j) \in E} x_t^{(i)\top} \mathbf{M}_\star x_t^{(j)} = \sum_{(i,j) \in E} z_t^{(i,j)\top} \theta_\star = \left\langle \sum_{(i,j) \in E} z_t^{(i,j)}, \theta_\star \right\rangle$

---

**Algorithm 1:** Adaptation of OFUL algorithm for Graphical Bilinear Bandit

---
**Input** : graph $\mathcal{G} = (V, E)$, node-arm set $\mathcal{X}$
$(V_1, V_2) = \text{Approx-MAX-CUT}(\mathcal{G})$
**for** $t = 1$ *to* $T$ **do**
    // Find the optimistic best couple of node-arms
    $\left(x_t, x_t', \tilde{\theta}_{t-1}\right) = \arg\max_{(x,x',\theta)\in\mathcal{X}^2 \times C_{t-1}} \langle z_{xx'} + z_{x'x}, \theta \rangle$;
    // Allocate $x_t$ and $x_t'$ in the graph
    $x_t^{(i)} = x_t$ for all $i$ in $V_1$;   $x_t^{(i)} = x_t'$ for all $i$ in $V_2$;
    Obtain for all $(i,j)$ in $E$ the rewards $y_t^{(i,j)}$;
    Compute $\hat{\theta}_t$ as in (3)
**end**
return $\hat{\theta}_t$

---

---

**Algorithm 2:** Approx-MAX-CUT

---
**Input** : $\mathcal{G} = (V, E)$
Set $V_1 = \emptyset$, $V_2 = \emptyset$
**for** $i$ *in* $V$ **do**
    $n_1 = \text{Card}\left(\{(i,j) \in E \mid j \in V_1\}\right)$;
    $n_2 = \text{Card}\left(\{(i,j) \in E \mid j \in V_2\}\right)$;
    **if** $n_1 > n_2$ **then** $V_2 \leftarrow V_2 \cup \{i\}$ **else** $V_1 \leftarrow V_1 \cup \{i\}$;
**end**
return $(V_1, V_2)$

---

we consider the couple of node-arms $(x_\star, x_\star') = \arg\max_{(x,x')\in\mathcal{X}^2} \langle z_{xx'} + z_{x'x}, \theta_\star \rangle$ maximizing the reward along one edge.[4] Given the optimal joint arm $(x_\star^{(1)}, \ldots, x_\star^{(n)})$ and the associated optimal edge-arms $z_\star^{(i,j)} = \text{vec}\left(x_\star^{(i)} x_\star^{(j)}\right)$ for all $(i, j) \in E$, we have

$$\langle z_\star^{(i,j)} + z_\star^{(j,i)}, \theta_\star \rangle \leq \langle z_{x_\star x_\star'} + z_{x_\star' x_\star}, \theta_\star \rangle \ . \tag{4}$$

Hence, an alternative objective is to construct as many edge-arms $z_{x_\star x_\star'}$ and $z_{x_\star' x_\star}$ as possible in the graph. Naturally, we do not have access to $\theta_\star$, so we use the principle of optimism in the face of uncertainty, which is to find the couple $(x_t, x_t')$ such that

$$(x_t, x_t') = \arg\max_{(x,x')\in\mathcal{X}^2} \max_{\theta \in C_{t-1}(\delta)} \langle z_{xx'} + z_{x'x}, \theta \rangle \ ,$$

and then allocate the node-arms to maximize the number of locally-optimal edge-arms $z_{x_t x_t'}$ and $z_{x_t' x_t}$. The motivation of this alternative goal can be understood thanks to (4) where we know that the associated expected reward is better than the one obtained with the edge-arms created with the optimal joint arms. Of course, one can also easily understand that for a general type of graph it is not possible to allocate those edge-arms everywhere (*i.e.,* to all the edges). For example take a complete graph of three nodes, allocating the edge-arms $z_{x_t x_t'}$ and $z_{x_t' x_t}$ is equivalent to allocate $x_t$ and $x_t'$ to the nodes. By doing so, two nodes will pull $x_t$ and the third one will pull $x_t'$, which makes it inevitable to allocate sub-optimal and unwanted edge-arms of the form $z_{x_t x_t}$.

The main concern with this method is to control how many unwanted edge-arms are drawn in each round (relative to the total number $m$ of edges) in order to minimize their impact on the resulting regret. Assigning $x_t$ to a subset of nodes and $x_t'$ to the complementary is equivalent to cutting the graph into two pieces and creating two distinct sets of nodes $V_1$ and $V_2$ such that $V = V_1 \cup V_2$ and $V_1 \cap V_2 = \emptyset$. Pulling the maximum amount of optimal edge-arms thus boils down to finding a cut passing through the maximum number of edges.

---

[4]The reward of an edge here denotes the sum of the rewards of the two corresponding directed edges in $E$.

This problem is known as the Max-Cut problem, which is also NP-Hard. However, the extensive attention this problem has received allows us to use one of the many approximation algorithms (see, *e.g.,* Algorithm 2) which are guaranteed to yield a cut passing through at least a given fraction of the edges in the graph.[5]

**Proposition 3.1.** *Given a graph $\mathcal{G} = (V, E)$, Algorithm 2 returns a couple $(V_1, V_2)$ such that*

$$\mathrm{Card}\left(\{(i,j) \in E \mid (i \in V_1 \wedge j \in V_2) \vee (i \in V_2 \wedge j \in V_1)\}\right) \geq \frac{m}{2} \ .$$

Using Algorithm 2 we can thus divide the set of nodes $V$ into two subsets $V_1$ and $V_2$ with the guarantee that at least $m/2$ edges connect the two subsets. Combining this algorithm with the principle of optimism in the face of uncertainty for linear bandits Abbasi-Yadkori et al. [2011] we obtain Algorithm 1 for which we can state the following guarantees on the $\alpha$-regret.

**Theorem 3.2.** *Given the Graphical Bilinear Bandit problem defined in Section 2, let $0 \leq \gamma \leq 1$ be a problem-dependent parameter defined by*

$$\gamma = \min_{x \in \mathcal{X}} \frac{\langle z_{xx}, \theta_\star \rangle}{\frac{1}{m} \sum_{(i,j) \in E} \left\langle z_\star^{(i,j)}, \theta_\star \right\rangle} \geq 0 \ ,$$

*and set $\alpha = \frac{1+\gamma}{2}$, then the $\alpha$-regret of Algorithm 1 satisfies*

$$R_\alpha(T) \leq \tilde{O}\left(\left(\sigma d^2 + S\sqrt{\lambda}\right)\sqrt{Tm \max\left(2, (LS)^2\right)}\right) + LSm\left\lceil d^2 \log_2\left(\frac{TmL^2/\lambda}{\delta}\right)\right\rceil \ ,$$

*where $\tilde{O}$ hides the logarithmic factors.*

One can notice that the first term of the regret-bound matches the one of a standard linear bandit that pulls sequentially $Tm$ edge-arms. The second term captures the cost of parallelizing $m$ draws of edge-arms per round. Indeed, the intuition behind this term is that the couple $(x_t, x'_t)$ chosen at round $t$ (and thus after having already pulled $tm$ edge-arms and received $tm$ rewards) is relevant to pull the $(tm + 1)$-th edge-arm but not necessarily the $(m - 1)$ edge-arms that follows (from the $(tm + 2)$-th to the $(tm + m)$-th one) since the reward associated to the $(tm + 1)$-th edge-arms could have led to change the central entity choice if it had been done sequentially. In Chan et al. [2021], they characterize this phenomenon and show that this potential change in choice occurs less and less often as we pull arms and get rewards, hence the dependence in $O(\log(Tm))$.

*What is $\gamma$ and what value can we expect?* This parameter measures what minimum gain with respect to the optimal reward one could get by constructing the edge-arms of the form $z_{xx}$. For example, if there exists $x_0 \in \mathcal{X}$ such that $\langle z_{x_0 x_0}, \theta_\star \rangle = 0$, then $\gamma = 0$ as well and we are in the worst case scenario where we can only have a guarantee on an $\alpha$-regret with $\alpha = 1/2$. In practice, having $\gamma = 0$ is reached when given the couple $(x_\star, x'_\star) = \arg\max_{(x,x') \in \mathcal{X}^2} \langle z_{xx'} + z_{x'x}, \theta_\star \rangle$, this $x_0$ is either $x_\star$ or $x'_\star$. In other words, if the unwanted edge-arms associated to the couple $(x_\star, x'_\star)$ gives low rewards, then the guarantee on the regret will be badly impacted. Hence we can wonder how we can prevent this phenomenon. We answer this question in the next section by taking into account both the proportion of undesirable edge arms allocated and their potential rewards at the selection of the pair $(x_t, x'_t)$ in order to improve in practice but also theoretically the performance of the algorithm.

## 4 Improved Algorithm for Graphical Bilinear Bandits

In this section, we want to capitalize on Algorithm 1 and its $\frac{1+\gamma}{2}$-no-regret property to optimize the allocation of the arms $x_t$ and $x'_t$ such that the unwanted and suboptimal arms $z_{x_t x_t}$ and $z_{x'_t x'_t}$ penalize as less as possible the reward.

---

[5]Most of the guarantees for the approximation of the Max-Cut problem are stated with respect to the optimal Max-Cut solution, which is not exactly the guarantee we are looking for: we need a guarantee as a proportion of the total number of edges. We thus have to be careful on the algorithm we choose.

Indeed, in Algorithm 1, the choice of the couple $(x_t, x_t')$ is driven by the optimistic reward of the edge-arms $z_{x_t x_t'}$ and $z_{x_t x_t'}$ but not on the one of the unwanted arms $z_{x_t x_t}$ and $z_{x_t' x_t'}$. Here, the first improvement that we can provide is to include them in the selection of the couple $(x_t, x_t')$ as follows

$$(x_t, x_t') = \arg\max_{(x,x')\in\mathcal{X}^2} \max_{\theta\in C_{t-1}(\delta)} \langle z_{xx'} + z_{x'x} + z_{xx} + z_{x'x'}, \theta \rangle \ .$$

However, this selection suggests that the number of unwanted edge-arms $z_{x_t x_t}$ and $z_{x_t' x_t'}$ will be equal to the number of optimal edge-arm $z_{x_t x_t'}$ and $z_{x_t' x_t}$ (in the sense of optimism), which is in general not true.[6] Hence, one has to take into consideration the proportion of edges that will be allocated by suboptimal and unwanted edge-arms or by optimistically optimal edge-arms.

To do so, let us define $m_1$ (respectively $m_2$) the number of edges that goes from nodes in $V_1$ (respectively $V_2$) to nodes in $V_1$ as well (respectively $V_2$) and $m_{1\to 2}$ (respectively $m_{2\to 1}$) the number of edges that goes from nodes in $V_1$ (respectively $V_2$) to nodes in $V_2$ (respectively $V_1$). One can notice that by definition of the edge set $E$, we have $m_{1\to 2} = m_{2\to 1}$ and that the total number of edges $m = m_{1\to 2} + m_{2\to 1} + m_1 + m_2$. By choosing the couple $(x_t, x_t')$ as follows

$$(x_t, x_t') = \arg\max_{(x,x')\in\mathcal{X}^2} \max_{\theta\in C_{t-1}(\delta)} \langle m_{1\to 2}\cdot z_{xx'} + m_{2\to 1}\cdot z_{x'x} + m_1\cdot z_{xx} + m_2\cdot z_{x'x'}, \theta \rangle \ ,$$

we are optimizing the total optimistic reward that one would obtain when allocating only two arms $(x, x') \in \mathcal{X}^2$ in the graph. This strategy is described in Algorithm 3.

---

**Algorithm 3:** Improved OFUL for Graphical Bilinear Bandits

**Input** : graph $\mathcal{G} = (V, E)$, node-arm set $\mathcal{X}$
$(V_1, V_2) = \text{Approx-MAX-CUT}(\mathcal{G})$;
$m_{1\to 2} = m_{2\to 1} = \text{Card}\left(\{(i,j)\in E | i\in V_1 \wedge j\in V_2\}\right)$;
$m_1 = \text{Card}\left(\{(i,j)\in E | i\in V_1 \wedge j\in V_1\}\right)$;  $m_2 = \text{Card}\left(\{(i,j)\in E | i\in V_2 \wedge j\in V_2\}\right)$;
**for** $t = 1$ *to* $T$ **do**
$\quad \left(x_t, x_t', \tilde{\theta}_{t-1}\right) =$
$\quad \arg\max_{(x,x',\theta)\in\mathcal{X}^2\times C_{t-1}} \langle m_{1\to 2}\cdot z_{xx'} + m_{2\to 1}\cdot z_{x'x} + m_1\cdot z_{xx} + m_2\cdot z_{x'x'}, \theta\rangle$;
$\quad x_t^{(i)} = x_t$ for all $i$ in $V_1$;  $x_t^{(i)} = x_t'$ for all $i$ in $V_2$;
$\quad$ Obtain for all $(i,j)$ in $E$ the rewards $y_t^{(i,j)}$;
$\quad$ Compute $\hat{\theta}_t$ as in (3)
**end**
return $\hat{\theta}_t$

---

Before stating the guarantees on the $\alpha$-regret of this improved algorithm, let us recall that we defined $(x_\star, x_\star') = \arg\max_{(x,x')\in\mathcal{X}} \langle z_{xx'} + z_{x'x}, \theta_\star \rangle$ and let us define the couple $(\tilde{x}_\star, \tilde{x}_\star')$ such that

$$(\tilde{x}_\star, \tilde{x}_\star') = \arg\max_{(x,x')\in\mathcal{X}} \langle m_{1\to 2}\cdot z_{xx'} + m_{2\to 1}\cdot z_{x'x} + m_1\cdot z_{xx} + m_2\cdot z_{x'x'}, \theta_\star \rangle \ .$$

We define $\Delta \geq 0$ to be the expected reward difference of allocating $(\tilde{x}_\star, \tilde{x}_\star')$ instead of $(x_\star, x_\star')$,

$$\Delta = \langle m_{1\to 2}\left(z_{\tilde{x}_\star \tilde{x}_\star'} - z_{x_\star x_\star'}\right) + m_{2\to 1}\left(z_{\tilde{x}_\star' \tilde{x}_\star} - z_{x_\star' x_\star}\right)$$
$$+ m_1\left(z_{\tilde{x}_\star \tilde{x}_\star} - z_{x_\star x_\star}\right) + m_2\left(z_{\tilde{x}_\star' \tilde{x}_\star'} - z_{x_\star' x_\star'}\right), \theta_\star \rangle \ .$$

The new guarantees that we get on the $\alpha$-regret of Algorithm 3 are stated in the following theorem.

---

[6]It is for instance true in complete graphs with an even number of nodes $n$ but it is already false when the number of nodes become odd.

Table 1: Value of several parameters with respect to the type of graph. Experiments were performed on graphs of $n = 100$ nodes, and results for the random graph are averaged over 100 draws.

| | Graph types | | | | |
| --- | --- | --- | --- | --- | --- |
| | Complete | Random | Circle | Star | Matching |
| $\frac{m1+m2}{m}$ | 0.495 | 0.453 | 0.01 | 0 | 0 |
| $\alpha_1$ | $0.5 + 0.5\gamma$ | | | | |
| $\alpha_2$ | $0.505 + 0.495\gamma + \epsilon$ | $0.547 + 0.453\gamma + \epsilon$ | $0.99 + 0.01\gamma + \epsilon$ | 1 | 1 |

**Theorem 4.1.** *Given the Graphical Bilinear Bandit problem defined as in Section 2, let $\gamma$ be defined as in Theorem 3.2, let $0 \leq \epsilon \leq \frac{1}{2}$ be a problem dependent parameter that measures the gain of optimizing on the suboptimal and unwanted arms defined as:*

$$\epsilon = \frac{\Delta}{\sum_{(i,j)\in E}\langle z_\star^{(i,j)}, \theta_\star\rangle} \ ,$$

*and set $\alpha = 1 - \left[\frac{m1+m2}{m}(1-\gamma) - \epsilon\right]$ where $\alpha \geq 1/2$ by construction, then the $\alpha$-regret of Algorithm 3 satisfies*

$$R_\alpha(T) \leq \tilde{O}\left(\left(\sigma d^2 + S\sqrt{\lambda}\right)\sqrt{Tm\max\left(2, (LS)^2\right)}\right) + LSm\left[d^2\log_2\left(\frac{TmL^2/\lambda}{\delta}\right)\right] \ ,$$

*where $\tilde{O}$ hides the logarithmic factors.*

Here one can see that the improvement happens in the $\alpha$ of the $\alpha$-regret. Let us analyze more precisely this term. First of all, notice that in a complete graph (which is the worst case scenario in terms of graph type and guarantees of the cut), we get $\alpha \geq \frac{1+\gamma}{2} + \epsilon$ which shows that optimizing over the suboptimal arms already improves our bound by $\epsilon$. On the contrary, in the most favorable graphs, which are bipartite graphs (*i.e.,* graphs where all the $m$ edges goes from nodes in $V_1$ to nodes in $V_2$ or vice versa), we have $m_2 + m_1 = 0$ and $\epsilon = 0$ which gives $\alpha = 1$ and makes the Algorithm 3 a no-regret algorithm. What may also be of interest is to understand how $\alpha$ varies with respect to $\gamma$, $\epsilon$ and the quantity $m_1$ and $m_2$ for graphs that are between a complete graph and a bipartite graph. We investigate experimentally this dependency in Section 5.

## 5 Experiments - Influence of the problem parameters on the regret

In this section, we give some insights on the problem-dependent parameters $\gamma$ and $\epsilon$ and the corresponding $\alpha$. Let $\alpha_1$ and $\alpha_2$ be the $\alpha$ stated respectively in Theorem 3.2 and Theorem 4.1. In the first experiment, we show the dependence of $\alpha_1$ and $\alpha_2$ on the graph type and the chosen approximation algorithm for the max-cut problem with respect to $\gamma$ and $\epsilon$. We also highlight the differences between the two parameters $\alpha_1$ and $\alpha_2$ and the significant improvement in guarantees that one can obtain using Algorithm 3 depending on the type of the graph. The results are presented in Table 1.

One can notice that the complete graph seems to be the one that gives the worst guarantee on the $\alpha$-regret with respect to $\epsilon$ and $\gamma$. Thus, we conducted a second experiment where we consider the worst case scenario in terms of the graph type –*e.g.,* the complete graph– and where there is $n = 10$ agents. The second experiment studies the variation of $\epsilon$ and $\gamma$ with respect to the unknown parameter matrix $\mathbf{M}_\star$. To design such experiments, we consider the arm-set $\mathcal{X}$ as the vectors $(e_1, \ldots, e_d)$ of canonical base in $\mathbb{R}^d$, which implies by construction that the arm-set $\mathcal{Z}$ contains the vectors $(e_1, \ldots, e_{d^2})$ of the canonical base in $\mathbb{R}^{d^2}$. We generate the matrix $\mathbf{M}_\star$ randomly in the following way: first, all elements of the matrix are drawn i.i.d. from a standard normal distribution, and then we take the absolute value of each of these elements to ensure that the matrix only contains positive numbers. The choice of the vectors of the canonical base as the arm-set allows us to modify the matrix $\mathbf{M}_\star$ and to illustrate in a simple way the dependence on $\gamma$ and $\epsilon$. Consider the best couple $(i^\star, j^\star) = \arg\max_{(i,j)\in\{1,\ldots,d\}^2}\langle z_{e_ie_j} + z_{e_je_i}, \theta_\star\rangle$, we want to see how the reward of the suboptimal

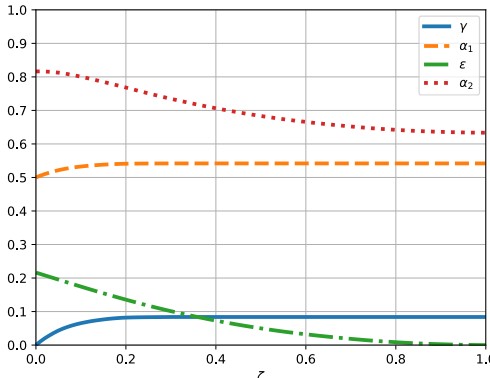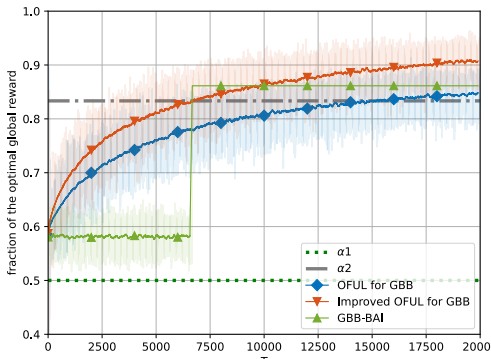

Figure 1: Variation of $\epsilon$, $\gamma$, $\alpha_1$ and $\alpha_2$ with respect to the parameter $\zeta$. The closer $\zeta$ is to 0 the lower the reward of the unwanted arms $z_{e_{i^\star}e_{i^\star}}$ and $z_{e_{i^\star}e_{i^\star}}$, the closer $\zeta$ is to 1 the higher the reward of the unwanted arms. The dimension $d$ of the arm-set is 10 (which gives linear reward with unknown parameter $\theta_\star$ of dimension 100). The plotted curve represents the average value of the parameters over 100 different matrices $\mathbf{M}_\star$ initiated randomly with positive values.

Figure 2: Fraction of the global reward obtained at each round by applying the Algorithm 1, Algorithm 3 and the Explore-Then-commit algorithm (here named GBB-BAI) using the exploration strategy in Rizk et al. [2021]. We use a complete graph of 5 nodes, we run the experiment on 5 different matrices as in Figure 1 with $\zeta = 0$ and run it 10 different times to plot the average fraction of the global reward

edge-arms $z_{e_{i^\star}e_{i^\star}}$ and $z_{e_{j^\star}e_{j^\star}}$ impact the value of $\gamma$, $\epsilon$ and thus $\alpha$. Notice that the reward associated to the edge-arm $z_{e_{i^\star}e_{i^\star}}$ (respectively $z_{e_{j^\star}e_{j^\star}}$) is $\mathbf{M}_{\star i^\star i^\star}$ (respectively $\mathbf{M}_{\star j^\star j^\star}$). Hence we define $0 \leq \zeta < 1$ and set $\mathbf{M}_{\star i^\star i^\star} = \mathbf{M}_{\star j^\star j^\star} = \zeta \times \frac{1}{2}(\mathbf{M}_{\star i^\star j^\star} + \mathbf{M}_{\star j^\star i^\star})$. We study the variation of $\gamma$, $\epsilon$, $\alpha_1$ and $\alpha_2$ with respect to $\zeta$. The results are presented in Figure 1. One can see that when the associated rewards of $z_{e_{i^\star}e_{i^\star}}$ and $z_{e_{j^\star}e_{j^\star}}$ are low (thus $\gamma$ is low and $\epsilon$ high), Algorithm 3 gives a much better guarantees than Algorithm 1 since it will focus on other node-arms than $e_{i^\star}$ and $e_{j^\star}$ that will give a higher global reward. Moreover, even when the unwanted edge-arm gives a high reward, the guarantees on the regret of Algorithm 3 are still stronger because it takes into consideration the quantities $m_1$ and $m_2$ of the constructed suboptimal edge-arms.

Finally, we design a last experiment that compares in practice the performance of Algorithm 1 and Algorithm 3 with the Explore-Then-Commit algorithm by using the exploration strategy designed in Rizk et al. [2021] during the exploration phase, and by allocating the nodes in $V_1$ and $V_2$ with the best estimated couple $(x, x') = \arg\max_{(x,x')}\langle z_{xx'} + z_{x'x}, \hat{\theta}_t\rangle$ during the commit phase.

## 6   Conclusion & discussions

In this paper, we presented the first regret-based algorithm for the stochastic graphical bilinear bandit problem with guarantees on the $\alpha$-regret. We showed that by exploiting the graph structure and the typology of the problem, one can both improve the performance in practice and have a better theoretical guarantee on its $\alpha$-regret. Also, we showed experimentally that our algorithm achieves a better performance than Explore-Then-Exploit on our synthetic datasets. The method presented in this article can be extended in many ways. First, one can consider cutting the graph into 3 or more pieces, which is equivalent to approximating the problem of a *Max-k-Cut* [Frieze and Jerrum, 1997] with $k \geq 3$. With the knowledge of such a partition of nodes $V_1, \ldots, V_k$, one may want to look for a $k$-tuple of node-arms maximizing the optimistic allocated reward rather than a pair, therefore introducing an elegant tradeoff between the optimality of the solution and the computational complexity of the arms allocation. One could also study this problem in the adversarial setting, in particular adapting adversarial linear bandit algorithms to our case. Finally, our setting could be extended to the case where each agent has its own reward matrix.

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
