# A  Proof of Proposition 3.1

*Proof.* Let consider the subgraph $\mathcal{G}^{(i)}$ containing all the nodes that have been assigned to $V_1$ or $V_2$ at the end of iteration $i$ of Algorithm 2. Let us denote $m^{(i)}$ the number of edges in the graph $\mathcal{G}^{(i)}$.

At the first iteration, the algorithm chooses the node 1, computes $n_1 = 0$ and $n_2 = 0$, and then assigns node 1 to $V_1$. With only one node in $\mathcal{G}^{(1)}$, we have $m^{(1)} = 0$. By denoting $c^{(i)}$ the number of additional cut edges induces by the assignment of node $i$ at iteration $i$, we have

$$\sum_{i=1}^{1} c^{(i)} = c^{(1)} = 0 \geq \frac{m^{(1)}}{2} \tag{5}$$

Indeed, at the end of iteration 1, there is only one node assigned, hence the number of cut edges induced by this assignment is $c^{(1)} = 0$.

Suppose that $\sum_{i=1}^{p} c^{(i)} \geq \frac{m^{(p)}}{2}$ for a certain $p \in \{1, \ldots, n-1\}$, let us prove that $\sum_{i=1}^{p+1} c^{(i)} \geq m^{(p+1)}/2$.

Indeed, at the iteration $p+1$, the algorithm chooses the node $(p+1)$ and computes $n_1$ and $n_2$. Since $n_1$ represents the number of neighbors of the node $(p+1)$ in $V_1$, if the node $p+1$ is added to $V_2$, then $2 \times n_1$ edges would be cut (the factor 2 comes from the fact that between two nodes $i$ and $j$, there are the edges $(i,j)$ and $(j,i)$). Similarly, since $n_2$ represents the number of neighbors of the node $(p+1)$ in $V_2$, if the node $(p+1)$ is added to $V_1$, then $2 \times n_2$ edges would be cut. Notice also that there is a total of $2 \times n_1 + 2 \times n_2$ edges between the node $(p+1)$ and the nodes in $\mathcal{G}^{(p)}$. In the algorithm, the node $(p+1)$ is added to $V_1$ or $V_2$ such that we cut the most edges, indeed one has

$$c^{(p+1)} = \max\left(2n_1, 2n_2\right) \geq \frac{2n_1 + 2n_2}{2} = n_1 + n_2 \ .$$

Hence,

$$\sum_{i=1}^{p+1} c^{(i)} = \sum_{i=1}^{p} c^{(i)} + c^{(p+1)} \geq \frac{m^{(p)}}{2} + c^{(p+1)} \geq \frac{m^{(p)}}{2} + n_1 + n_2 \tag{6}$$

The number of edges that is added to the subgraph $\mathcal{G}^{(p)}$ when adding the node $(p+1)$ is equal to $2n_1 + 2n_2 = m^{(p+1)} - m^{(p)}$, hence,

$$\frac{m^{(p)}}{2} + n_1 + n_2 = \frac{m^{(p)}}{2} + \frac{m^{(p+1)} - m^{(p)}}{2} = \frac{m^{(p+1)}}{2} \tag{7}$$

We have shown that $\sum_{i=1}^{1} c^{(i)} \geq \frac{m^{(1)}}{2}$ and that if $\sum_{i=1}^{p} c^{(i)} \geq \frac{m^{(p)}}{2}$ for a certain $p \in \{1, \ldots, n-1\}$, then $\sum_{i=1}^{p+1} c^{(i)} \geq \frac{m^{(p+1)}}{2}$. Thus, $\sum_{i=1}^{p} c^{(i)} \geq \frac{m^{(p)}}{2}$ for any $p \in \{1, \ldots, n\}$, especially for $p = n$ where $\mathcal{G}^{(n)} = \mathcal{G}$. By definition $\sum_{i=1}^{n} c^{(i)}$ is the total number of edges that are cut which also means that

$$\sum_{i=1}^{n} c^{(i)} = \text{Card}\left\{(i,j) \in E \mid (i \in V_1 \wedge j \in V_2) \vee (i \in V_2 \wedge j \in V_1)\right\} \ .$$

$\square$

# B  Proof of Theorem 3.2 and Theorem 4.1

To properly derive the regret bounds, we will have to make connections between our setting and a standard linear bandit that chooses sequentially $Tm$ arms. For that matter, let us consider an arbitrary

order on the set of edges $E$ and denote $E[i]$ the $i$-th edge according to this order with $i \in \{1, \ldots, m\}$. We define for all $t \in \{1, \ldots, T\}$ and $p \in \{1, \ldots, m\}$ the OLS estimator

$$\hat{\theta}_{t,p} = \mathbf{A}_{t,p}^{-1} b_{t,p} \ ,$$

where

$$\mathbf{A}_{t,p} = \lambda \mathbf{I}_{d^2} + \sum_{s=1}^{t-1}\sum_{b=1}^{m} z_s^{E[b]} z_s^{E[b]\top} + \sum_{k=1}^{p} z_t^{E[k]} z_t^{E[k]\top} \ ,$$

with $\lambda$ a regularization parameter and

$$b_{t,p} = \sum_{s=1}^{t-1}\sum_{b=1}^{m} z_s^{E[b]} y_s^{E[b]} + \sum_{k=1}^{p} z_t^{E[k]} y_t^{E[k]} \ . \tag{8}$$

We define also the confidence set

$$C_{t,p}(\delta) = \left\{ \theta : \|\theta - \hat{\theta}_{t,p}\|_{A_{t,p}^{-1}} \leq \sigma \sqrt{d^2 \log\left(\frac{1 + tmL^2/\lambda}{\delta}\right)} + \sqrt{\lambda} S \right\} \ , \tag{9}$$

where with probability $1 - \delta$, we have that $\theta_\star \in C_{t,p}(\delta)$ for all $t \in \{1, \ldots, T\}$, $p \in \{1, \ldots, m\}$ and $\delta \in (0, 1]$.

Notice that the confidence set $C_t(\delta)$ defined in Section 3 is exactly the confidence set $C_{t,m}(\delta)$ defined here. The definitions of the matrix $A_{t,m}$ and the vector $b_{t,m}$ follow the same reasoning.

## B.1 Proof of Theorem 3.2

*Proof.* Recall that $(x_\star^{(1)}, \ldots, x_\star^{(n)}) = \arg\max_{(x^{(1)}, \ldots, x^{(n)})} \sum_{(i,j) \in E} x^{(i)\top} \mathbf{M}_\star x^{(j)}$ is the optimal joint arm, and we define for each edge $(i,j) \in E$ the optimal edge arm $z_\star^{(i,j)} = \text{vec}(x_\star^{(i)} x_\star^{(j)\top})$.

We recall that the $\alpha$-pseudo-regret is

$$R_\alpha(T) \triangleq \sum_{t=1}^{T} \sum_{(i,j) \in E} \alpha \langle z_\star^{(i,j)}, \theta_\star \rangle - \langle z_t^{(i,j)}, \theta_\star \rangle \tag{10}$$

$$= R(T) - \sum_{t=1}^{T} \sum_{(i,j) \in E} (1 - \alpha) \langle z_\star^{(i,j)}, \theta_\star \rangle \ , \tag{11}$$

where the pseudo-regret $R(T)$ is defined by

$$R(T) = \sum_{t=1}^{T} \sum_{(i,j) \in E} \langle z_\star^{(i,j)}, \theta_\star \rangle - \langle z_t^{(i,j)}, \theta_\star \rangle \ .$$

Let us borrow the notion of *Critical Covariance Inequality* introduced in [Chan et al., 2021], that is for a given round $t \in \{1, \ldots, T\}$ and $p \in \{1, \ldots, m\}$, the expected covariance matrix $\mathbf{A}_{t,p}$ satisfies the critical covariance inequality if

$$\mathbf{A}_{t-1,m} \preccurlyeq \mathbf{A}_{t,p} \preccurlyeq 2\mathbf{A}_{t-1,m} \ . \tag{12}$$

Let us now define the event $D_t$ as the event where at a given round $t$, for all $p \in \{1, \ldots, m\}$, $\mathbf{A}_{t,p}$ satisfies the critical covariance inequality (CCI).

We can write the pseudo-regret as follows:

$$R(T) = \sum_{t=1}^{T} \mathbb{1}[D_t] \sum_{(i,j) \in E} \langle z_\star^{(i,j)}, \theta_\star \rangle - \left\langle z_t^{(i,j)}, \theta_\star \right\rangle + \mathbb{1}[D_t^c] \sum_{(i,j) \in E} \langle z_\star^{(i,j)}, \theta_\star \rangle - \left\langle z_t^{(i,j)}, \theta_\star \right\rangle$$

$$\leq \underbrace{\sum_{t=1}^{T} \mathbb{1}[D_t] \sum_{(i,j) \in E} \langle z_\star^{(i,j)}, \theta_\star \rangle - \left\langle z_t^{(i,j)}, \theta_\star \right\rangle}_{(a)} + \underbrace{LSm \sum_{t=1}^{T} \mathbb{1}[D_t^c]}_{(b)} \ .$$

We know that the approximation Max-CUT algorithm returns two subsets of nodes $V_1$ and $V_2$ such that there are at least $m/2$ edges between $V_1$ and $V_2$, and to be more precise: at least $m/4$ edges from $V_1$ to $V_2$ and at least $m/4$ edges from $V_2$ to $V_1$. Hence at each time $t$, if all the nodes of $V_1$ pull the node-arm $x_t$ and all the nodes of $V_2$ pull the node-arm $x_t'$, we can derive the term $(a)$ as follows:

$$(a) = \sum_{t=1}^{T} \mathbb{1}[D_t] \sum_{(i,j) \in E} \langle z_\star^{(i,j)}, \theta_\star \rangle - \langle z_t^{(i,j)}, \theta_\star \rangle$$

$$= \sum_{t=1}^{T} \mathbb{1}[D_t] \sum_{(i,j) \in E} \langle z_\star^{(i,j)}, \theta_\star \rangle - \mathbb{1}\left[i \in V_1 \wedge j \in V_2\right] \langle z_t^{(i,j)}, \theta_\star \rangle$$

$$- \mathbb{1}\left[i \in V_2 \wedge j \in V_1\right] \langle z_t^{(i,j)}, \theta_\star \rangle$$

$$- \mathbb{1}\left[i \in V_1 \wedge j \in V_1\right] \langle z_t^{(i,j)}, \theta_\star \rangle$$

$$- \mathbb{1}\left[i \in V_2 \wedge j \in V_2\right] \langle z_t^{(i,j)}, \theta_\star \rangle \ .$$

Notice that $\sum_{(i,j) \in E} z_\star^{(i,j)} = \sum_{(i,j) \in E} \frac{1}{m} \sum_{(k,l) \in E} z_\star^{(k,l)}$, so one has

$$(a) = \underbrace{\sum_{t=1}^{T} \mathbb{1}[D_t] \sum_{(i,j) \in E} \mathbb{1}\left[i \in V_1 \wedge j \in V_2\right] \left( \left\langle \frac{1}{m} \sum_{(k,l) \in E} z_\star^{(k,l)}, \theta_\star \right\rangle - \langle z_t^{(i,j)}, \theta_\star \rangle \right)}_{(a_1)}$$

$$+ \underbrace{\sum_{t=1}^{T} \mathbb{1}[D_t] \sum_{(i,j) \in E} \mathbb{1}\left[i \in V_2 \wedge j \in V_1\right] \left( \left\langle \frac{1}{m} \sum_{(k,l) \in E} z_\star^{(k,l)}, \theta_\star \right\rangle - \langle z_t^{(i,j)}, \theta_\star \rangle \right)}_{(a_2)}$$

$$+ \underbrace{\sum_{t=1}^{T} \mathbb{1}[D_t] \sum_{(i,j) \in E} \mathbb{1}\left[i \in V_1 \wedge j \in V_1\right] \left( \left\langle \frac{1}{m} \sum_{(k,l) \in E} z_\star^{(k,l)}, \theta_\star \right\rangle - \langle z_t^{(i,j)}, \theta_\star \rangle \right)}_{(a_3)}$$

$$+ \underbrace{\sum_{t=1}^{T} \mathbb{1}[D_t] \sum_{(i,j) \in E} \mathbb{1}\left[i \in V_2 \wedge j \in V_2\right] \left( \left\langle \frac{1}{m} \sum_{(k,l) \in E} z_\star^{(k,l)}, \theta_\star \right\rangle - \langle z_t^{(i,j)}, \theta_\star \rangle \right)}_{(a_4)} \ .$$

Let us analyse the first term:

$$(a_1) = \sum_{t=1}^{T} \mathbb{1}[D_t] \sum_{i=1}^{n} \sum_{\substack{j \in N_i \\ j>i}} \mathbb{1}\left[i \in V_1 \wedge j \in V_2\right] \left\langle \frac{2}{m} \sum_{(k,l) \in E} z_\star^{(k,l)} - \left( z_t^{(i,j)} + z_t^{(j,i)} \right), \theta_\star \right\rangle \ . \quad (13)$$

By defining $(x_\star, x'_\star) = \arg\max_{(x,x') \in \mathcal{X}^2} \langle z_{xx'} + z_{x'x}, \theta_\star \rangle$, and noticing that in the case where a node $i$ is in $V_1$ and a neighbouring node $j$ in is $V_2$, then $z_t^{(i,j)} = z_{x_t x'_t}$, we have,

$$
\begin{aligned}
\frac{2}{m} \sum_{(k,l) \in E} \left\langle z_\star^{(k,l)}, \theta_\star \right\rangle &= \frac{2}{m} \sum_{k=1}^n \sum_{\substack{j \in \mathcal{N}_k \\ j>k}} \left\langle z_\star^{(k,l)} + z_\star^{(l,k)}, \theta_\star \right\rangle \\
&\leq \frac{2}{m} \sum_{k=1}^n \sum_{\substack{j \in \mathcal{N}_k \\ j>k}} \langle z_{x_\star x'_\star} + z_{x'_\star x_\star}, \theta_\star \rangle \\
&= \langle z_{x_\star x'_\star} + z_{x'_\star x_\star}, \theta_\star \rangle \\
&\leq \langle z_{x_t x'_t} + z_{x'_t x_t}, \tilde{\theta}_{t-1,m} \rangle \qquad \text{w.p } 1-\delta \\
&= \langle z_t^{(i,j)} + z_t^{(j,i)}, \tilde{\theta}_{t-1,m} \rangle \ .
\end{aligned}
$$

Plugging this last inequality in (13) yields, with probability $1-\delta$,

$$
\begin{aligned}
(a_1) &\leq \sum_{t=1}^T \mathbb{1}[D_t] \sum_{i=1}^n \sum_{\substack{j \in \mathcal{N}_i \\ j>i}} \mathbb{1}\left[ i \in V_1 \wedge j \in V_2 \right] \left\langle z_t^{(i,j)} + z_t^{(j,i)}, \tilde{\theta}_{t-1,m} - \theta_\star \right\rangle \\
&= \sum_{t=1}^T \mathbb{1}[D_t] \sum_{(i,j) \in E} \mathbb{1}\left[ i \in V_1 \wedge j \in V_2 \right] \left\langle z_t^{(i,j)}, \tilde{\theta}_{t-1,m} - \theta_\star \right\rangle \ .
\end{aligned}
$$

We define, as in Algorithm 1, $\mathbb{1}\left[ z_t^{(i,j)} = z_{x_t x'_t} \right] \triangleq \mathbb{1}\left[ i \in V_1 \wedge j \in V_2 \right]$. Then, one has, with probability $1-\delta$,

$$
\begin{aligned}
(a_1) &\leq \sum_{t=1}^T \mathbb{1}[D_t] \sum_{(i,j) \in E} \mathbb{1}\left[ z_t^{(i,j)} = z_{x_t x'_t} \right] \left\langle z_t^{(i,j)}, \tilde{\theta}_{t-1,m} - \theta_\star \right\rangle \\
&= \sum_{t=1}^T \mathbb{1}[D_t] \sum_{k=1}^m \mathbb{1}\left[ z_t^{E[k]} = z_{x_t x'_t} \right] \left\langle z_t^{E[k]}, \tilde{\theta}_{t-1,m} - \theta_\star \right\rangle \\
&= \sum_{t=1}^T \mathbb{1}[D_t] \sum_{k=1}^m \mathbb{1}\left[ z_t^{E[k]} = z_{x_t x'_t} \right] \left\langle z_t^{E[k]}, \tilde{\theta}_{t-1,m} - \hat{\theta}_{t-1,m} \right\rangle + \left\langle z_t^{E[k]}, \hat{\theta}_{t-1,m} - \theta_\star \right\rangle \\
&\leq \sum_{t=1}^T \mathbb{1}[D_t] \sum_{k=1}^m \mathbb{1}\left[ z_t^{E[k]} = z_{x_t x'_t} \right] \|z_t^{E[k]}\|_{\mathbf{A}_{t,k-1}^{-1}} \|\tilde{\theta}_{t-1,m} - \hat{\theta}_{t-1,m}\|_{\mathbf{A}_{t,k-1}} \\
&\qquad + \mathbb{1}\left[ z_t^{E[k]} = z_{x_t x'_t} \right] \|z_t^{E[k]}\|_{\mathbf{A}_{t,k-1}^{-1}} \|\hat{\theta}_{t-1,m} - \theta_\star\|_{\mathbf{A}_{t,k-1}} \\
&\leq \sum_{t=1}^T \mathbb{1}[D_t] \sum_{k=1}^m \mathbb{1}\left[ z_t^{E[k]} = z_{x_t x'_t} \right] \|z_t^{E[k]}\|_{\mathbf{A}_{t,k-1}^{-1}} \sqrt{2} \|\tilde{\theta}_{t-1,m} - \hat{\theta}_{t-1,m}\|_{\mathbf{A}_{t-1,m}} \qquad (14) \\
&\qquad + \mathbb{1}\left[ z_t^{E[k]} = z_{x_t x'_t} \right] \|z_t^{E[k]}\|_{\mathbf{A}_{t,k-1}^{-1}} \sqrt{2} \|\hat{\theta}_{t-1,m} - \theta_\star\|_{\mathbf{A}_{t-1,m}} \\
&\leq \sum_{t=1}^T \sum_{k=1}^m \mathbb{1}\left[ z_t^{E[k]} = z_{x_t x'_t} \right] 2\sqrt{2\beta_t(\delta)} \|z_t^{E[k]}\|_{\mathbf{A}_{t,k-1}^{-1}} \qquad (15) \\
&\leq \sum_{t=1}^T \sum_{k=1}^m 2\sqrt{2\beta_t(\delta)} \|z_t^{E[k]}\|_{\mathbf{A}_{t,k-1}^{-1}} \ , \qquad (16)
\end{aligned}
$$

with $\sqrt{\beta_t(\delta)} \leq \sigma \sqrt{d^2 \log\left(\frac{1+tmL^2/\lambda}{\delta}\right)} + \sqrt{\lambda}S$ and where (14) uses the critical covariance inequality (12), (15) comes from the definition of the confidence set $C_{t-1,m}(\delta)$ (9) and (16) upper bounds the indicator functions by 1.

Using a similar reasoning, we obtain the same bound for $(a_2)$:

$$(a_2) \leq \sum_{t=1}^{T}\sum_{k=1}^{m} 2\sqrt{2\beta_t(\delta)}\|z_t^{E[k]}\|_{\mathbf{A}_{t,k-1}^{-1}} \quad . \tag{17}$$

Let us bound the terms $(a_3)$ and $(a_4)$.

$$(a_3) = \sum_{t=1}^{T} \mathbb{1}[D_t] \sum_{(i,j)\in E} \mathbb{1}\left[z_t^{(i,j)} = z_{x_t x_t}\right]\left(\left\langle \frac{1}{m}\sum_{(k,l)\in E} z_\star^{(k,l)}, \theta_\star \right\rangle - \left\langle z_t^{(i,j)}, \theta_\star \right\rangle\right) \tag{18}$$

For all $x \in \mathcal{X}$, let $\gamma_x$ be the following ratio

$$\gamma_x = \frac{\langle z_{xx}, \theta_\star \rangle}{\left\langle \frac{1}{m}\sum_{(k,l)\in E} z_\star^{(k,l)}, \theta_\star \right\rangle} \quad , \tag{19}$$

and let $\gamma$ be the worst ratio

$$\gamma = \min_{x\in\mathcal{X}} \frac{\langle z_{xx}, \theta_\star \rangle}{\left\langle \frac{1}{m}\sum_{(k,l)\in E} z_\star^{(k,l)}, \theta_\star \right\rangle} \quad . \tag{20}$$

We have

$$\begin{aligned}
(a_3) &= \sum_{t=1}^{T} \mathbb{1}[D_t] \sum_{(i,j)\in E} \mathbb{1}\left[z_t^{(i,j)} = z_{x_t x_t}\right]\left(\left\langle \frac{1}{m}\sum_{(k,l)\in E} z_\star^{(k,l)}, \theta_\star \right\rangle - \gamma_{x_t}\left\langle \frac{1}{m}\sum_{(k,l)\in E} z_\star^{(k,l)}, \theta_\star \right\rangle\right) \\
&\leq \sum_{t=1}^{T} \mathbb{1}[D_t] \sum_{(i,j)\in E} \mathbb{1}\left[z_t^{(i,j)} = z_{x_t x_t}\right]\left(\left\langle \frac{1}{m}\sum_{(k,l)\in E} z_\star^{(k,l)}, \theta_\star \right\rangle - \gamma\left\langle \frac{1}{m}\sum_{(k,l)\in E} z_\star^{(k,l)}, \theta_\star \right\rangle\right) \\
&= \sum_{t=1}^{T} \mathbb{1}[D_t] \sum_{(i,j)\in E} \mathbb{1}\left[z_t^{(i,j)} = z_{x_t x_t}\right](1-\gamma)\left\langle \frac{1}{m}\sum_{(k,l)\in E} z_\star^{(k,l)}, \theta_\star \right\rangle \\
&\leq T\frac{m}{4}(1-\gamma)\left\langle \frac{1}{m}\sum_{(k,l)\in E} z_\star^{(k,l)}, \theta_\star \right\rangle \tag{21} \\
&= \sum_{t=1}^{T} \sum_{(i,j)\in E} \frac{1}{4}(1-\gamma)\left\langle z_\star^{(i,j)}, \theta_\star \right\rangle \quad ,
\end{aligned}$$

where (21) comes from the fact that there is at most $m/4$ edges that goes from node in $V_1$ to other nodes in $V_1$ and that $\mathbb{1}[D_t] \leq 1$ for all $t$.

The derivation of this bound for $(a_3)$ gives the same one for $(a_4)$

$$(a_4) \leq \sum_{t=1}^{T} \sum_{(i,j)\in E} \frac{1}{4}(1-\gamma)\left\langle z_\star^{(i,j)}, \theta_\star \right\rangle \quad . \tag{22}$$

By rewriting $(a)$, we have :

$$(a) \leq \sum_{t=1}^{T} \sum_{k=1}^{m} 4\sqrt{2\beta_t(\delta)} \|z_t^{E[k]}\|_{\mathbf{A}_{t,k-1}^{-1}} + \frac{1}{2}(1-\gamma)\langle z_\star^{(i,j)}, \theta_\star \rangle \ .$$

In [Chan et al., 2021], they bounded the term $(b)$ as follows

$$LSm \sum_{t=1}^{T} \mathbb{1}[D_t^c] \leq LSm \left\lceil d^2 \log_2 \left( \frac{TmL^2/\lambda}{\delta} \right) \right\rceil \ . \tag{23}$$

We thus have the regret bounded by

$$R(T) \leq \sum_{t=1}^{T} \sum_{k=1}^{m} 4\sqrt{2\beta_t(\delta)} \|z_t^{E[k]}\|_{\mathbf{A}_{t,k-1}^{-1}} + \frac{1}{2}(1-\gamma)\langle z_\star^{(i,j)}, \theta_\star \rangle + LSm \left\lceil d^2 \log_2 \left( \frac{TmL^2/\lambda}{\delta} \right) \right\rceil \ ,$$

which gives us

$$R_{\frac{1+\gamma}{2}}(T) \leq \sum_{t=1}^{T} \sum_{k=1}^{m} 4\sqrt{2\beta_t(\delta)} \|z_t^{E[k]}\|_{\mathbf{A}_{t,k-1}^{-1}} + LSm \left\lceil d^2 \log_2 \left( \frac{TmL^2/\lambda}{\delta} \right) \right\rceil \ .$$

Let us bound the first term with the double sum as it is done in [Abbasi-Yadkori et al., 2011, Chan et al., 2021]:

$$\sum_{t=1}^{T} \sum_{k=1}^{m} 4\sqrt{2\beta_t(\delta)} \|z_t^{E[k]}\|_{\mathbf{A}_{t,k-1}^{-1}}$$

$$\leq \sum_{t=1}^{T} \sum_{k=1}^{m} \min\left(2LS, 4\sqrt{2\beta_t(\delta)} \|z_t^{E[k]}\|_{\mathbf{A}_{t,k-1}^{-1}}\right)$$

$$\leq \sum_{t=1}^{T} \sum_{k=1}^{m} 4\sqrt{2\beta_t(\delta)} \min\left(LS, \|z_t^{E[k]}\|_{\mathbf{A}_{t,k-1}^{-1}}\right)$$

$$\leq \sqrt{Tm \times 32\beta_T(\delta) \sum_{t=1}^{T} \sum_{k=1}^{m} \min\left((LS)^2, \|z_t^{E[k]}\|_{\mathbf{A}_{t,k-1}^{-1}}^2\right)}$$

$$\leq \sqrt{32Tm\beta_T(\delta) \sum_{t=1}^{T} \sum_{k=1}^{m} \max\left(2, (LS)^2\right) \log\left(1 + \|z_t^{E[k]}\|_{\mathbf{A}_{t,k-1}^{-1}}^2\right)} \tag{24}$$

$$= \sqrt{32Tm\beta_T(\delta) \max\left(2, (LS)^2\right) \sum_{t=1}^{T} \sum_{k=1}^{m} \log\left(1 + \|z_t^{E[k]}\|_{\mathbf{A}_{t,k-1}^{-1}}^2\right)}$$

$$\leq \sqrt{32Tm\beta_T(\delta) \max\left(2, (LS)^2\right) d^2 \log\left(1 + \frac{TmL^2/\lambda}{d^2}\right)} \tag{25}$$

$$\leq \sqrt{32Tmd^2 \max\left(2, (LS)^2\right) \log\left(1 + \frac{TmL^2/\lambda}{d^2}\right)} \left(\sigma\sqrt{d^2 \log\left(\frac{1 + TmL^2/\lambda}{\delta}\right)} + \sqrt{\lambda}S\right)$$

where (24) uses the fact that for all $a, x \geq 0$, $\min(a, x) \leq \max(2, a) \log(1 + x)$, (25) uses the fact that $\sum_{t=1}^{T} \sum_{k=1}^{m} \log \left( 1 + \|z_t^{E[k]}\|_{\mathbf{A}_{t,k-1}^{-1}}^2 \right) \leq d^2 \log \left( 1 + \frac{TmL^2/\lambda}{d^2} \right)$ from Lemma 19.4 in Lattimore and Szepesvári [2018].

The final bound for the $\frac{1+\gamma}{2}$-regret is

$$R_{\frac{1+\gamma}{2}}(T) \leq \sqrt{32Tmd^2 \max\left(2, (LS)^2\right) \log\left(1 + \frac{TmL^2/\lambda}{d^2}\right)} \left( \sigma \sqrt{d^2 \log\left(\frac{1 + TmL^2/\lambda}{\delta}\right)} + \sqrt{\lambda}S \right)$$
$$+ LSm \left\lceil d^2 \log_2 \left( \frac{TmL^2/\lambda}{\delta} \right) \right\rceil$$

$\square$

## B.2   Proof of Theorem 4.1

*Proof.* For the sake of completeness in the proof we recall that we defined the couples $(x_\star, x_\star')$ and $(\tilde{x}_\star, \tilde{x}_\star')$ and the quantity $\Delta$ as follows:

$$(x_\star, x_\star') = \underset{(x,x')\in\mathcal{X}^2}{\arg\max} \langle z_{xx'} + z_{x'x}, \theta_\star \rangle$$

$$(\tilde{x}_\star, \tilde{x}_\star') = \underset{(x,x')\in\mathcal{X}}{\arg\max} \langle m_{1\to2} \cdot z_{xx'} + m_{2\to1} \cdot z_{x'x} + m_1 \cdot z_{xx} + m_2 \cdot z_{x'x'}, \theta_\star \rangle .$$

and

$$\Delta = \langle m_{1\to2} \left( z_{\tilde{x}_\star \tilde{x}_\star'} - z_{x_\star x_\star'} \right) + m_{2\to1} \left( z_{\tilde{x}_\star' \tilde{x}_\star} - z_{x_\star' x_\star} \right)$$
$$+ m_1 \left( z_{\tilde{x}_\star \tilde{x}_\star} - z_{x_\star x_\star} \right) + m_2 \left( z_{\tilde{x}_\star' \tilde{x}_\star'} - z_{x_\star' x_\star'} \right), \theta_\star \rangle .$$

And we recall that in Algorithm 3, the tuple $(x_t, x_t', \tilde{\theta}_{t-1,m})$ is obtained as follows:

$$\left( x_t, x_t', \tilde{\theta}_{t-1,m} \right) = \underset{(x,x',\theta)\in\mathcal{X}^2\times C_{t-1}}{\arg\max} \langle m_{1\to2} \cdot z_{xx'} + m_{2\to1} \cdot z_{x'x} + m_1 \cdot z_{xx} + m_2 \cdot z_{x'x'}, \theta \rangle$$

We can write the regret $R(T)$ as in the proof of Theorem 3.2:

$$R(T) = \sum_{t=1}^{T} \mathbb{1}[D_t] \sum_{(i,j)\in E} \langle z_\star^{(i,j)}, \theta_\star \rangle - \left\langle z_t^{(i,j)}, \theta_\star \right\rangle + \mathbb{1}[D_t^c] \sum_{(i,j)\in E} \langle z_\star^{(i,j)}, \theta_\star \rangle - \left\langle z_t^{(i,j)}, \theta_\star \right\rangle$$

$$\leq \underbrace{\sum_{t=1}^{T} \mathbb{1}[D_t] \sum_{(i,j)\in E} \langle z_\star^{(i,j)}, \theta_\star \rangle - \left\langle z_t^{(i,j)}, \theta_\star \right\rangle}_{(a)} + \underbrace{LSm \sum_{t=1}^{T} \mathbb{1}[D_t^c]}_{(b)}$$

Here, $(b)$ doesn't change, we thus only focus on deriving $(a)$.

$$(a) = \sum_{t=1}^{T} \mathbb{1}[D_t] \sum_{(i,j)\in E} \langle z_\star^{(i,j)}, \theta_\star \rangle - \langle z_t^{(i,j)}, \theta_\star \rangle$$

$$\leq \sum_{t=1}^{T} \sum_{(i,j)\in E} \langle z_\star^{(i,j)}, \theta_\star \rangle - \langle z_t^{(i,j)}, \theta_\star \rangle \qquad \text{(where } \mathbb{1}[D_t] \leq 1\text{)}$$

$$= \underbrace{\sum_{t=1}^{T} \sum_{(i,j)\in E} \frac{m_{1\to2} + m_{2\to1}}{m} \langle z_\star^{(i,j)}, \theta_\star \rangle}_{(a_1)} + \sum_{t=1}^{T} \sum_{(i,j)\in E} \frac{m_1 + m_2}{m} \langle z_\star^{(i,j)}, \theta_\star \rangle - \sum_{t=1}^{T} \sum_{(i,j)\in E} \langle z_t^{(i,j)}, \theta_\star \rangle$$

We have

$$(a_1) = \sum_{t=1}^{T} \sum_{(i,j) \in E} \frac{2m_{1 \to 2}}{m} \langle z_\star^{(i,j)}, \theta_\star \rangle$$

$$= \sum_{t=1}^{T} \sum_{i=1}^{n} \sum_{\substack{j \in \mathcal{N}_i \\ j > i}} \frac{2m_{1 \to 2}}{m} \langle z_\star^{(i,j)} + z_\star^{(j,i)}, \theta_\star \rangle$$

$$\leq \sum_{t=1}^{T} \sum_{i=1}^{n} \sum_{\substack{j \in \mathcal{N}_i \\ j > i}} \frac{2m_{1 \to 2}}{m} \langle z_{x_\star x_\star'} + z_{x_\star' x_\star}, \theta_\star \rangle$$

$$= \sum_{t=1}^{T} \sum_{i=1}^{n} \sum_{\substack{j \in \mathcal{N}_i \\ j > i}} \frac{2}{m} \langle m_{1 \to 2} \cdot z_{x_\star x_\star'} + m_{2 \to 1} \cdot z_{x_\star' x_\star}, \theta_\star \rangle$$

$$= \sum_{t=1}^{T} \sum_{i=1}^{n} \sum_{\substack{j \in \mathcal{N}_i \\ j > i}} \frac{2}{m} \langle m_{1 \to 2} \cdot z_{x_\star x_\star'} + m_{2 \to 1} \cdot z_{x_\star' x_\star} + m_1 \cdot z_{x_\star x_\star} + m_2 \cdot z_{x_\star' x_\star'}, \theta_\star \rangle$$

$$\quad - \sum_{t=1}^{T} \sum_{i=1}^{n} \sum_{\substack{j \in \mathcal{N}_i \\ j > i}} \frac{2}{m} \langle m_1 \cdot z_{x_\star x_\star} + m_2 \cdot z_{x_\star' x_\star'}, \theta_\star \rangle$$

$$= \sum_{t=1}^{T} \sum_{i=1}^{n} \sum_{\substack{j \in \mathcal{N}_i \\ j > i}} \frac{2}{m} \langle m_{1 \to 2} \cdot z_{\tilde{x}_\star \tilde{x}_\star'} + m_{2 \to 1} \cdot z_{\tilde{x}_\star' \tilde{x}_\star} + m_1 \cdot z_{\tilde{x}_\star \tilde{x}_\star} + m_2 \cdot z_{\tilde{x}_\star' \tilde{x}_\star'}, \theta_\star \rangle - \frac{2}{m} \Delta$$

$$\quad - \sum_{t=1}^{T} \sum_{i=1}^{n} \sum_{\substack{j \in \mathcal{N}_i \\ j > i}} \frac{2}{m} \langle m_1 \cdot z_{x_\star x_\star} + m_2 \cdot z_{x_\star' x_\star'}, \theta_\star \rangle$$

$$= \sum_{t=1}^{T} \langle m_{1 \to 2} \cdot z_{\tilde{x}_\star \tilde{x}_\star'} + m_{2 \to 1} \cdot z_{\tilde{x}_\star' \tilde{x}_\star} + m_1 \cdot z_{\tilde{x}_\star \tilde{x}_\star} + m_2 \cdot z_{\tilde{x}_\star' \tilde{x}_\star'}, \theta_\star \rangle - \Delta$$

$$\quad - \sum_{t=1}^{T} \langle m_1 \cdot z_{x_\star x_\star} + m_2 \cdot z_{x_\star' x_\star'}, \theta_\star \rangle$$

$$\leq \sum_{t=1}^{T} \left\langle m_{1 \to 2} \cdot z_{x_t x_t'} + m_{2 \to 1} \cdot z_{x_t' x_t} + m_1 \cdot z_{x_t x_t} + m_2 \cdot z_{x_t' x_t'}, \tilde{\theta}_{t-1,m} \right\rangle - \Delta \qquad \text{w.p } 1 - \delta$$

$$\quad - \sum_{t=1}^{T} \langle m_1 \cdot z_{x_\star x_\star} + m_2 \cdot z_{x_\star' x_\star'}, \theta_\star \rangle$$

$$= \sum_{t=1}^{T} \sum_{(i,j) \in E} \langle z_t^{(i,j)}, \tilde{\theta}_{t-1,m} \rangle - \sum_{t=1}^{T} \Delta - \sum_{t=1}^{T} \langle m_1 \cdot z_{x_\star x_\star} + m_2 \cdot z_{x_\star' x_\star'}, \theta_\star \rangle$$

By plugging the last upper bound in $(a)$ and with probability $1 - \delta$, we have,

$$(a) \leq \sum_{t=1}^{T} \sum_{(i,j)\in E} \langle z_t^{(i,j)}, \tilde{\theta}_{t-1,m} \rangle - \sum_{t=1}^{T} \Delta - \sum_{t=1}^{T} \langle m_1 \cdot z_{x_\star x_\star} + m_2 \cdot z_{x'_\star x'_\star}, \theta_\star \rangle$$

$$+ \sum_{t=1}^{T} \sum_{(i,j)\in E} \frac{m_1 + m_2}{m} \langle z_\star^{(i,j)}, \theta_\star \rangle - \sum_{t=1}^{T} \sum_{(i,j)\in E} \langle z_t^{(i,j)}, \theta_\star \rangle$$

$$= \sum_{t=1}^{T} \sum_{(i,j)\in E} \langle z_t^{(i,j)}, \tilde{\theta}_{t-1,m} - \theta_\star \rangle - \sum_{t=1}^{T} \Delta - \sum_{t=1}^{T} \langle m_1 \cdot z_{x_\star x_\star} + m_2 \cdot z_{x'_\star x'_\star}, \theta_\star \rangle$$

$$+ \sum_{t=1}^{T} \sum_{(i,j)\in E} \frac{m_1 + m_2}{m} \langle z_\star^{(i,j)}, \theta_\star \rangle$$

$$= \sum_{t=1}^{T} \sum_{(i,j)\in E} \langle z_t^{(i,j)}, \tilde{\theta}_{t-1,m} - \theta_\star \rangle - \sum_{t=1}^{T} \Delta - \sum_{t=1}^{T} \sum_{(i,j)\in E} \frac{m_1}{m}\gamma_{x_\star}\langle z_\star^{(i,j)}, \theta_\star \rangle + \frac{m_2}{m}\gamma_{x'_\star}\langle z_\star^{(i,j)}, \theta_\star \rangle$$

$$+ \sum_{t=1}^{T} \sum_{(i,j)\in E} \frac{m_1 + m_2}{m} \langle z_\star^{(i,j)}, \theta_\star \rangle$$

$$\leq \sum_{t=1}^{T} \sum_{(i,j)\in E} \langle z_t^{(i,j)}, \tilde{\theta}_{t-1,m} - \theta_\star \rangle - \sum_{t=1}^{T} \Delta - \sum_{t=1}^{T} \sum_{(i,j)\in E} \frac{m_1 + m_2}{m}\gamma\langle z_\star^{(i,j)}, \theta_\star \rangle + \sum_{t=1}^{T} \sum_{(i,j)\in E} \frac{m_1 + m_2}{m} \langle z_\star^{(i,j)}, \theta_\star \rangle$$

$$= \sum_{t=1}^{T} \sum_{(i,j)\in E} \langle z_t^{(i,j)}, \tilde{\theta}_{t-1,m} - \theta_\star \rangle - \sum_{t=1}^{T} \Delta + \sum_{t=1}^{T} \sum_{(i,j)\in E} \frac{m_1 + m_2}{m}(1-\gamma)\langle z_\star^{(i,j)}, \theta_\star \rangle$$

$$= \sum_{t=1}^{T} \sum_{(i,j)\in E} \langle z_t^{(i,j)}, \tilde{\theta}_{t-1,m} - \theta_\star \rangle - \sum_{t=1}^{T} \sum_{(i,j)\in E} \epsilon \langle z_\star^{(i,j)}, \theta_\star \rangle + \sum_{t=1}^{T} \sum_{(i,j)\in E} \frac{m_1 + m_2}{m}(1-\gamma)\langle z_\star^{(i,j)}, \theta_\star \rangle$$

$$= \sum_{t=1}^{T} \sum_{(i,j)\in E} \langle z_t^{(i,j)}, \tilde{\theta}_{t-1,m} - \theta_\star \rangle + \sum_{t=1}^{T} \sum_{(i,j)\in E} \left[\frac{m_1 + m_2}{m}(1-\gamma) - \epsilon\right] \langle z_\star^{(i,j)}, \theta_\star \rangle$$

By plugging $(a)$ in the regret and with probability $1 - \delta$, we have,

$$R(T) \leq \sum_{t=1}^{T} \sum_{(i,j)\in E} \langle z_t^{(i,j)}, \tilde{\theta}_{t-1,m} - \theta_\star \rangle + \sum_{t=1}^{T} \sum_{(i,j)\in E} \left[\frac{m_1 + m_2}{m}(1-\gamma) - \epsilon\right] \langle z_\star^{(i,j)}, \theta_\star \rangle + LSm\sum_{t=1}^{T} \mathbb{1}[D_t^c]$$

which gives,

$$R(T) - \sum_{t=1}^{T} \sum_{(i,j)\in E} \left[\frac{m_1 + m_2}{m}(1-\gamma) - \epsilon\right] \langle z_\star^{(i,j)}, \theta_\star \rangle \leq \sum_{t=1}^{T} \sum_{(i,j)\in E} \langle z_t^{(i,j)}, \tilde{\theta}_{t-1,m} - \theta_\star \rangle + LSm\sum_{t=1}^{T} \mathbb{1}[D_t^c]$$

$$R_{1-\left[\frac{m_1+m_2}{m}(1-\gamma)-\epsilon\right]}(T) \leq \sum_{t=1}^{T} \sum_{(i,j)\in E} \langle z_t^{(i,j)}, \tilde{\theta}_{t-1,m} - \theta_\star \rangle + LSm\sum_{t=1}^{T} \mathbb{1}[D_t^c]$$

The upper bound of the right hand term follows exactly what we have already done for Theorem 3.2 by applying the upper bounds (16) and (23) ◻

# C  Additional information on the experiments

## C.1  Table 1

The number of nodes in each graph is equal to 100. The random graph corresponds to a graph where for two nodes $i$ and $j$ in $V$, the probability that $(i, j)$ and $(j, i)$ is in $E$ is equal to 0.6. The results for the random graph are averaged over 100 draws. The matching graph represents the graph where each node $i \in V$ has only one neighbour: $\mathrm{Card}(\mathcal{N}_i) = 1$.

## C.2  Figure 1

The graph used in this experiment is a complete graph of 10 nodes. The arm set $\mathcal{X} = \{e_1, \ldots, e_d\}$ which gives $\mathcal{Z} = \{e_1, \ldots, e_{d^2}\}$. The matrix $\mathbf{M}_\star$ is randomly initialized such that all elements of the matrix are drawn i.i.d. from a standard normal distribution, and then we take the absolute value of each of these elements to ensure that the matrix only contains positive numbers. We plotted the results by varying $\zeta$ from 0 to 1 with a step of 0.01. We conducted the experiment on 100 different matrices $\mathbf{M}_\star$ randomly initialized as explained above and plotted the average value of the obtained $\gamma$, $\epsilon$, $\alpha_1$ and $\alpha_2$.

## C.3  Figure 2

For the last experiment, we used a complete graph of 5 nodes. The arm set $\mathcal{X} = \{e_1, \ldots, e_d\}$ which gives $\mathcal{Z} = \{e_1, \ldots, e_{d^2}\}$. The matrix $\mathbf{M}_\star$ is randomly initialized as explained in the previous experiment. We fixed $\zeta = 0$ and the horizon $T = 20000$. We ran the experiment 10 times and plotted the average values (shaded curve) and the moving average curve with a window of 100 steps for more clarity.

The Explore-Then-Commit algorithm has an exploration phase of $T/3$ rounds and then exploits by pulling the couple $(x_t, x_t') = \arg\max_{(x, x')} \langle z_{xx'} + z_{x'x}, \hat{\theta}_t \rangle$. Note that we set the exploration phase to $T/3$ because most of the time, it was sufficient for the learner to have the estimated optimal pair $(x_t, x_t')$ equal to the real optimal pair $(x_\star, x_\star')$.

**Machine used for all the experiments** : Macbook Pro, Apple M1 chip, 8-core CPU

**The code is available** here.