# OpenReview forum: "An $\alpha$-No-Regret Algorithm For Graphical Bilinear Bandits"
_NeurIPS.cc/2022/Conference — NeurIPS 2022 Accept_

### Official Review · Reviewer_shTc · 2022-07-08

**Rating:** 7
**Confidence:** 4
**Soundness:** 3 good
**Presentation:** 4 excellent
**Contribution:** 3 good

**Summary:**

This paper addresses the problem of bilinear bandits with graphical structure. In this problem, the agent must attribute at each round $t\leq T$ an action $x_t^{(i)}$ for each node $i$ of a graph. Then, he receives an expected reward $\sum_{(i,j) \in E} x_t^{(i)} M_* x_t^{(j)}$, where $M_*$ is an unknown matrix, and E is the set of edges of the graph.

Even in the case where $M^*$ is known, the problem of finding the best action is known to be NP-hard. For this reason, the authors focus on the problem of minimizing the $\alpha$-regret for some $\alpha \in [1/2,1]$, i.e. the difference between $\alpha$ times the cumulative reward of the oracle agent, and their reward. They present two algorithms for this problem, and establish bounds on their $\alpha$-regret. They also study its empirical properties on simulated datasets.

**Questions:**

Can the author please either include a proof of Proposition 3.1 (regarding the efficiency of Max-Cut approximation), or provide a reference proving it?

Could you provide a reference more explicit for lower-bounding the probability of the critical covariance inequality? For the sake of completeness, it would be interesting to include a (sketch of) proof of this result.

I believe there might be a typo in the definition of $\epsilon$ in Theorem 4.1. It should be defined as $\frac{\Delta}{\sum \langle z_*^{(i,j)} \vert \theta_*\rangle}$, instead of $\frac{\Delta}{\sum z_*^{(i,j)}}$.

**Limitations:**

The authors discuss possible extensions of their work.

**Strengths And Weaknesses:**

Originality : This problem addresses a well motivated problem of bilinear bandits with a graph structure. While this problem is related to that of bilinear bandit, the parallelization changes the nature of the problem. This problem has already been studied with the objective of finding the best action, as well as in the linear setting, and some proof technics are common to these results.

Clarity : This paper is very well written, pedagogical, and the arguments are well explained.

Significance : This paper addresses a well-motivated problem that had not been studied previously.

Quality :  This paper is very well written, and addresses a problem not solved before. The numerical experiments are convincing.

---

> ### Author Response · Authors · 2022-08-02
> **Response to Reviewer shTc**
>
> Many thanks for your comments. We address them below.
>
> "Can the author please either include a proof of Proposition 3.1 (regarding the efficiency of Max-Cut approximation), or provide a reference proving it?"
>
> The proof which gives an approximation ratio of $1/2$ with respect to the max-cut value can be found in Sahni et al. 1976 "P-complete approximation problems". However, since we want an approximation ratio of the number of cut edges with respect to the total number of edges in the graph, we give the following proof, that we will add in the appendix for the sake of completeness:
>
> Proof of Proposition 3.1.
>
> Let us consider the subgraph $\mathcal{G}^{(i)}$ containing all the nodes that have been assigned to $V_1$ or $V_2$ at the end of iteration $i$. Let us denote $m^{(i)}$ the number of edges in the graph $\mathcal{G}^{(i)}$.
>
> At the first iteration, the algorithm chooses the node $1$, computes $n_1 = 0$ and $n_2 = 0$, and then assigns node $1$ to $V_1$. With only one node in $\mathcal{G}^{(1)}$, we have $m^{(1)}=0$. By denoting $c^{(i)}$ the number of additional cut edges induces by the assignment of node $i$ at iteration $i$, we have
>
> \begin{align}
>     \sum_{i=1}^{1} c^{(i)} = c^{(1)} = 0 \geq \frac{m^{(1)}}{2}
> \end{align}
>
> Indeed, at the end of iteration $1$, there is only one node assigned, hence the number of cut edges induced by this assignment is $c^{(1)}=0$.
>
> Suppose that $\sum_{i=1}^{p} c^{(i)} \geq  \frac{m^{(p)}}{2}$ for a certain $p \in \\{1,\dots, n-1\\}$, let us prove that $\sum_{i=1}^{p+1} c^{(i)} \geq  m^{(p+1)}/2$.
>
> Indeed, at the iteration $p+1$, the algorithm chooses the node $(p+1)$ and computes $n_1$ and $n_2$. Since $n_1$ represents the number of neighbors of the node $(p+1)$ in $V_1$, if the node $(p+1)$ is added to $V_2$, then $2 \times n_1$ edges would be cut (the factor $2$ comes from the fact that between two nodes $i$ and $j$, there are the edges $(i,j)$ and $(j,i)$). Similarly, since $n_2$ represents the number of neighbors of the node $(p+1)$ in $V_2$, if the node $(p+1)$ is added to $V_1$, then $2 \times n_2$ edges would be cut. Notice also that there is a total of $2 \times n_1 + 2 \times n_2$ edges between the node $(p+1)$ and the nodes in $G^{(p)}$.
> In the algorithm, the node $(p+1)$ is added to $V_1$ or $V_2$ such that we cut the most edges, indeed one has
>
> \begin{align*}
>     c^{(p+1)} = \max\left(2n_1, 2n_2\right) \geq \frac{2n_1 + 2n_2}{2} = n_1 + n_2\enspace.
> \end{align*}
>
> Hence,
>
> \begin{align}
>     \sum_{i=1}^{p+1} c^{(i)} = \sum_{i=1}^{p} c^{(i)} + c^{(p+1)} \geq \frac{m^{(p)}}{2} + c^{(p+1)} \geq  \frac{m^{(p)}}{2} + n_1 + n_2
> \end{align}
>
> The number of edges that is added to the subgraph $G^{(p)}$ when adding the node $(p+1)$ is equal to $2n_1 + 2n_2 = m^{(p+1)} - m^{(p)}$, hence,
>
> \begin{align}
>     \frac{m^{(p)}}{2} + n_1 + n_2 = \frac{m^{(p)}}{2} + \frac{m^{(p+1)} - m^{(p)}}{2} = \frac{m^{(p+1)}}{2}
> \end{align}
>
> We have shown that $\sum_{i=1}^{1} c^{(i)} \geq  \frac{m^{(1)}}{2}$ and that if $\sum_{i=1}^{p} c^{(i)} \geq  \frac{m^{(p)}}{2}$ for a certain $p \in \\{1,\dots, n-1\\}$, then $\sum_{i=1}^{p+1} c^{(i)} \geq  \frac{m^{(p+1)}}{2}$. Thus, $\sum_{i=1}^{p} c^{(i)} \geq  \frac{m^{(p)}}{2}$ for any  $p \in \\{1,\dots, n\\}$, especially for $p = n$ where $\mathcal{G}^{(n)} = \mathcal{G}$. By definition $\sum_{i=1}^{n} c^{(i)}$ is the total number of edges that are cut which also means that
> \begin{align}
> \sum_{i=1}^{n} c^{(i)} = \text{Card}(\\{(i,j) \in E \ | \ \left(i \in V_1 \wedge j \in V_2 \right) \vee \left(i \in V_2 \wedge j \in V_1 \right) \\}) \geq \frac{m}{2}
> \end{align}
>
> $\square$
>
>
> "Could you provide a reference more explicit for lower-bounding the probability of the critical covariance inequality? For the sake of completeness, it would be interesting to include a (sketch of) proof of this result."
>
> The reference is: Lemma 1. in Chan et al. 2021. "Parallelizing Contextual Linear Bandits".
> Let $D_t$ be the event that the critical covariance inequality is respected at time $t$, and $D^c_t$ its complementary.
> The authors show that
> \begin{align}
>     \sum_{t=1}^{T} \mathbb{1}[D^c_t] \leq \left\lceil d^2 \log_2\left(\frac{1+TmL^2}{d^2\lambda}\right)\right\rceil
> \end{align}
>
> "I believe there might be a typo in the definition of $\epsilon$ in Theorem 4.1. It should be defined as $\frac{\Delta}{\sum_{(i,j)\in E} \langle z_\star^{(i,j)} , \theta_\star \rangle}$, instead of $\frac{\Delta}{\sum_{(i,j)\in E}  z_\star^{(i,j)}}$"
>
> Thank you for pointing that out.

---

> > ### Comment · Reviewer_shTc · 2022-08-04
> > **Answer to the authors**
> >
> > I thank the authors for answering all my questions.

---

### Official Review · Reviewer_DZQi · 2022-07-13

**Rating:** 6
**Confidence:** 3
**Soundness:** 3 good
**Presentation:** 3 good
**Contribution:** 2 fair

**Summary:**

The paper claims to analyze the first regret-based approach to the Graphical Bilinear Bandits problem, where n agents in a graph play a stochastic bilinear bandit game with each of their neighbors. They showed that the setting reveals a combinatorial NP-hard problem that prevents the use of any existing regret-based algorithm in the (bi-)linear bandit literature. The authors present the first regret-based algorithm for graphical bilinear bandits using the principle of optimism in the face of uncertainty. Theoretical analysis of this new method yields an upper bound of the α-regret which characterizes the dependence of the graph structure on the rate of convergence. Experiments are provided in support of their claim.


**Questions:**

The O(d^2) dependency on regret is believed to be not tight in general. Can the authors comment on the tightness of the upper bound guarantee (for Thm 3.2 and 4.1)?
In the algorithmic ideas or regret analysis, is there a new idea that is specific to this setting distinct from what has been already explored for the linear bandit scenario?

**Limitations:**

See points for Weaknesses listed above.

**Strengths And Weaknesses:**

Strength:
- The alpha regret formulation and setting of bilinear bandit is well motivated since the original problem is NP-hard (Rizk et al,21).
- Theoretical regret guarantees
- Experimental evaluation

Weakness:
- Motivating applications are missing that fit the linearity framework
- Technical novelty: The proposed algorithmic idea posed the bilinear problem as a linear problem in d^2 dimensions to the best of my understanding. The remaining challenge is to lift the techniques from the d-dimension linear bandit problem and apply it to the current setting.
- Optimality guarantees are not tight?
- No comparisons with state-of-the-art bilinear optimization routines, e.g. As pointed out, we can pose the problem as a linear bandit problem in O(d^2) dimension, so in that case, I am curious to understand how the regret and runtime performance of the proposed methods compare against existing linear bandit algorithms, Thompson sampling or OFUL?

---

> ### Author Response · Authors · 2022-08-02
> **Response to Reviewer DZQi**
>
> Thank you for your review. We address your comments below.
>
> "Optimality guarantees are not tight?No comparisons with state-of-the-art bilinear optimization routines, e.g. As pointed out, we can pose the problem as a linear bandit problem in $O(d^2)$ dimension, so in that case, I am curious to understand how the regret and runtime performance of the proposed methods compare against existing linear bandit algorithms, Thompson sampling or OFUL?"
>
> Indeed, the graphical bilinear bandit can be presented as a global linear bandit with $K^n$ arms of dimension $d^2$: at each round $t$ the expected global reward obtained by the learner is of the form $\left\langle \sum_{(i,j)\in E} z_t^{(i,j)}, \theta_\star \right\rangle$, with $\sum_{(i,j)} z_t^{(i,j)}$ the arm chosen at time $t$. In the OFUL algorithm for example, the learner has to compute $K^n$ estimates of the reward to draw the optimistic arm, which is not tractable.
> This argument also works for Thompson sampling, where the learner first samples one $\theta$ parameter and then estimates the rewards of all $K^n$ global arms to decide which one to draw. It is therefore not possible to apply directly the existing algorithms in the linear bandit literature. In our paper, we address the problem of maximizing cumulative rewards in the graphical bilinear bandit model introduced by Rizk et al. 2021 where, to the best of our knowledge, there was no (polynomial time) algorithm for this regret-based approach yet. Our proposed algorithm can be considered as a first building block. Comparing our algorithm and the associated guarantees for graphical bilinear bandits (thus $n$ agents involving $m$ dependent (bi)-linear bandit problems) with an algorithm for a simple linear bandit in dimension $d^2$ (only one agent and thus only one bandit problem) is not so simple. The first obstacle to this comparison is that, if in a simple linear bandit model, a sublinear regret is possible, it is not the case here in a graphical bilinear bandit where one must rely on an $\alpha$-regret. Therefore, the comparison of the upper bounds of two different quantities seems a bit unfair and complicated. Nevertheless, note that in the upper bounds provided in Theorems 3.2 and 4.1, the first term reflects the same rate of convergence (for the $\alpha$-regret) as that of a simple linear bandit (for the regret) where $Tm$ arms are drawn sequentially. The second term is an additional cost due to parallelization when dealing with several agents.
>
>
> "The $O(d^2)$ dependency on regret is believed to be not tight in general. Can the authors comment on the tightness of the upper bound guarantee (for Thm 3.2 and 4.1)?"
>
> In the (bi-)linear bandit case, it is indeed not tight. In Jun et al. 2019 "Bilinear Bandits with low rank structure" presented a modified version of OFUL where for a single bilinear bandit gives an upper bound on the regret of $O((2d)^{3/2}\sqrt{rT})$. Doing this kind of assumption on the matrix is a serious perspective for future work.
>
> "In the algorithmic ideas or regret analysis, is there a new idea that is specific to this setting distinct from what has been already explored for the linear bandit scenario?"
>
> Yes, there is. In the graphical bilinear bandit, the first challenge was to deal with the NP-hardness of the problem. As stated in the paper, drawing the optimistic joint arm at each round $t$ is also NP-hard, which forces to rely on another objective. We proposed a relaxation that relies on a Max-Cut problem to find two node-arms that allow us to have a guarantee on the optimal joint arm. Note that even if $\mathbf{M}_\star$ were known to the learner, it would not be possible to simply use it to find the optimal joint arm because it is also NP-hard. This is not the case in a simple linear bandit where having the true parameter $\theta_\star$ directly gives the optimal arm in polynomial time. Another distinct idea from the linear bandit scenario is that at each iteration, the learner chooses suboptimal arms due to the structure of the graph that impacts the $\alpha$-regret, which we first highlight and then leverage to propose the improved version of the algorithm. The improvement is visible in the $\alpha$ of the $\alpha$-regret and we provide an analysis and experiments of the proposed $\alpha$ with respect to the graph and the matrix $\mathbf{M}_\star$ (which, in our opinion, is beyond the scope of the simple linear bandit scenario). Finally, while addressing these challenges, we still need to pay attention to the multi-agent and paralellism aspect of the problem, where the learner draws $m$ arms in each round.

---

### Official Review · Reviewer_7vV1 · 2022-07-22

**Rating:** 5
**Confidence:** 3
**Soundness:** 3 good
**Presentation:** 3 good
**Contribution:** 2 fair

**Summary:**

The paper describes an algorithm for achieving sublinear `\alpha-'regret for the problem of graphical bilinear bandits introduced by Rizk etal '21. The algorithm is based in the well-known optimism under uncertainty principle. Further they do some modification on some of the problem-dependent parameters to achieve better regret guarantees. They show some numerical results to support their claims.

**Questions:**

1) Regarding the analysis novelty: please see the comment above on technical novelty.

2) Since I am not very well versed on this topic, are their attempts to show any lower bounds for this problem setting? If not, is there a fundamental difficulty in showing such bounds?

**Limitations:**

I agree that the authors are transparent that the paper draws connections with existing works by Rizk etal and Abbasi yadkori etal 2011. However given these already existing works, the contributions made in this paper are not very surprising to me.

**Strengths And Weaknesses:**

+ The paper attempts to tackle the regret minimizing aspect of the graphical bilinear bandit problem, which is a positive contribution.
+ The paper is clearly written, defining the quantities of interest and citing appropriate literature works.
+ Experiments show their claims of changing the parameter values which support their claims

- The problem setting introduced is not novel, however this is not a major concern for me.
- The algorithms introduced are somewhat the first choice that one would follow for solving a regret minimization metric for this setting. It would be great if the authors make a small section on the technical challenges faced (and how they overcome them) while adapting the OFUL algorithm to their setting.

---

> ### Author Response · Authors · 2022-08-02
> **Response to Reviewer 7vV1**
>
> Many thanks for your comments. We address them below.
>
> "The algorithms introduced are somewhat the first choice that one would follow for solving a regret minimization metric for this setting. It would be great if the authors make a small section on the technical challenges faced (and how they overcome them) while adapting the OFUL algorithm to their setting. I agree that the authors are transparent that the paper draws connections with existing works by Rizk et al and Abbasi-Yadkori et al 2011. However given these already existing works, the contributions made in this paper are not very surprising to me."
>
> Thank you for your advice, we will definitely add a section on the technical challenges encountered when adapting the OFUL algorithm:
>
> We want to point out that the first choice one would make in adapting (or applying) the OFUL algorithm would be to choose at each round $t$ the optimistic joint arm $(x^{(1)}_t, \dots, x^{(n)}_t)$ such that
>
>
> \begin{align}(x_t^{(1)},\dots, x_t^{(n)}) = argmax_{(x^{(1)}, \dots, x^{(n)}) \in \mathcal{X}^n} \max_{\theta \in C_{t-1}(\delta)} \left\langle \sum_{(i,j)\in E} z_{x^{(i)} x^{(j)}},\theta \right\rangle\end{align}
>
> However, as explained in the article, this is an NP-Hard problem, and there is no polynomial time algorithm in $n$ that can return this joint arm.
>
> - The first challenge was therefore to rely on another optimization problem while keeping the idea of Optimism in the Face of Uncertainty (OFU). Although we retain this OFU principle as in Abbasi-Yadkori et al 2011, we face NP-Hard problems in tasks that were considered trivial in the original OFUL algorithm. In our paper, we choose to design a strategy based on the Max-Cut problem where we want to cut as many edges as possible and allocate the (optimistically) optimal edge-arm to the cut edges. Solving this underlying problem allows us to give a guarantee on the reward of the returned joint arm with respect to the optimal one. Note that the guarantees derived in the literature of the Max-Cut problem do not give the proportion of cut edges with respect to the total number of edges (which is what we want) but rather the approximate value of the maximum cut relative to the value of the max-cut (which is not of interest here). Moreover, using this kind of approximation algorithm also involves pulling suboptimal arms of the form $z_{xx}$ that impact these guarantees.
> - The second challenge was to understand how the suboptimal pulled arms impact the overall reward (which is outside the scope of both the Rizk et al. 2021 paper and the Abbasi-Yadkori et al. 2011 paper). We expressed the $\alpha$-regret in such a way that this impact of suboptimal arms is visible and measurable. We then propose a second algorithm based on this result that improves the choice of the chosen edge arm $z_{x_t x_t^\prime}$ in each round $t$, and also highlight this impact in the $\alpha$-regret. The theoretical guarantees have been explored experimentally to show precisely what can be expected from the structure of the graph or the unknown parameter $\mathbf{M}_\star$.
> - Finally, as mentioned in the paper, the multi-agent aspect involves dealing with the parallelization of decisions during a round. While in Abbasi-Yadkori et al 2011, this problem is not considered, and in Rizk et al 2021, they use random sampling for best-arm identification to avoid dealing with this problem, in the regret-based approach, one has to pay attention to the derivation of the regret (here in this framework, the $\alpha$-regret). As mentioned in the proofs, for that particular aspect we rely on the work of Chan et at. 2021 to derive the $\alpha$-regret correctly.
>
> Also, we believe it is of high interest to have analyzed and proposed this first algorithm that fills the gap in the literature and allow researcher to iterate on it (e.g, follow the perspective of using a max-k-cut approximation algorithm).
>
> "Since I am not very well versed on this topic, are their attempts to show any lower bounds for this problem setting? If not, is there a fundamental difficulty in showing such bounds?"
>
> Deriving a lower bound in this framework can be both simple and tricky. Indeed, while the graphical bilinear bandits can be seen as a global linear bandit of $K^n$ arms of dimension $d^2$, one can easily rely on existing lower bounds for the linear bandit. Note that the lower bound is relative to the regret, not the $\alpha$-regret. But it is understandable that having a lower bound that is unreachable by a polynomial-time algorithm is not really informative. Indeed, one may want to use the existing lower bounds for the linear bandit, but on the $\alpha$-regret. To be more precise, one may want to set an upper bound on $\alpha$ to assert that, asymptotically, an $\alpha$-regret of order $f(T)$ is unavoidable, where $\alpha$ cannot be larger than some quantity $\beta$, for any instance of graphical bilinear bandit. This work is highly non-trivial, and we leave it for future work.

---

### Meta-Review · Area_Chair_AsxM · 2022-08-20

**Recommendation:** Accept
**Confidence:** Certain

**Metareview:**

We thank the authors for their submission.

In this paper, agents are vertices on a graph where an edge between agents i,j indicates that they play a multi-armed bandit game against each other, where the expected reward is bilinear in both agents' actions. The goal is to maximize the total expected reward over all edges of the graph, and prior work shows that this is an NP-hard problem.
Contribution: efficient optimism-based algorithms for minimizing $\alpha$-approximate regret, thus circumventing the computational hardness.
Considering approximate regret in this setting is a novel approach.
The paper is clearly written and presents a convincing experimental evaluation.

**Award:**

No

---

### Decision · Program_Chairs · 2022-09-14

Accept